# The Total Variation on Hypergraphs - Learning on Hypergraphs Revisited

**Matthias Hein, Simon Setzer, Leonardo Jost and Syama Sundar Rangapuram**
Department of Computer Science
Saarland University

## Abstract

Hypergraphs allow one to encode higher-order relationships in data and are thus a very flexible modeling tool. Current learning methods are either based on approximations of the hypergraphs via graphs or on tensor methods which are only applicable under special conditions. In this paper, we present a new learning framework on hypergraphs which fully uses the hypergraph structure. The key element is a family of regularization functionals based on the total variation on hypergraphs.

## 1 Introduction

Graph-based learning is by now well established in machine learning and is the standard way to deal with data that encode pairwise relationships. Hypergraphs are a natural extension of graphs which allow to model also higher-order relations in data. It has been recognized in several application areas such as computer vision [1, 2], bioinformatics [3, 4] and information retrieval [5, 6] that such higher-order relations are available and help to improve the learning performance.

Current approaches in hypergraph-based learning can be divided into two categories. The first one uses tensor methods for clustering as the higher-order extension of matrix (spectral) methods for graphs [7, 8, 9]. While tensor methods are mathematically quite appealing, they are limited to so-called $k$-uniform hypergraphs, that is, each hyperedge contains exactly $k$ vertices. Thus, they are not able to model mixed higher-order relationships. The second main approach can deal with arbitrary hypergraphs [10, 11]. The basic idea of this line of work is to approximate the hypergraph via a standard weighted graph. In a second step, one then uses methods developed for graph-based clustering and semi-supervised learning. The two main ways of approximating the hypergraph by a standard graph are the clique and the star expansion which were compared in [12]. One can summarize [12] by stating that no approximation fully encodes the hypergraph structure. Earlier, [13] have proven that an exact representation of the hypergraph via a graph retaining its cut properties is impossible.

In this paper, we overcome the limitations of both existing approaches. For both clustering and semi-supervised learning the key element, either explicitly or implicitly, is the cut functional. Our aim is to directly work with the cut defined on the hypergraph. We discuss in detail the differences of the hypergraph cut and the cut induced by the clique and star expansion in Section 2.1. Then, in Section 2.2, we introduce the total variation on a hypergraph as the Lovasz extension of the hypergraph cut. Based on this, we propose a family of regularization functionals which interpolate between the total variation and a regularization functional enforcing smoother functions on the hypergraph corresponding to Laplacian-type regularization on graphs. They are the key for the semi-supervised learning method introduced in Section 3. In Section 4, we show in line of recent research [14, 15, 16, 17] that there exists a tight relaxation of the normalized hypergraph cut. In both learning problems, convex optimization problems have to be solved for which we derive scalable methods in Section 5. The main ingredients of these algorithms are proximal mappings for which we provide a novel algorithm and analyze its complexity. In the experimental section 6, we show that fully incorporating hypergraph structure is beneficial. *All proofs are moved to the supplementary material.*

## 2 The Total Variation on Hypergraphs

A large class of graph-based algorithms in semi-supervised learning and clustering is based either explicitly or implicitly on the cut. Thus, we discuss first in Section 2.1 the hypergraph cut and the corresponding approximations. In Section 2.2, we introduce in analogy to graphs, the total variation on hypergraphs as the Lovasz extension of the hypergraph cut.

### 2.1 Hypergraphs, Graphs and Cuts

Hypergraphs allow modeling relations which are not only pairwise as in graphs but involve multiple vertices. In this paper, we consider weighted undirected hypergraphs $H = (V, E, w)$ where $V$ is the vertex set with $|V| = n$ and $E$ the set of hyperedges with $|E| = m$. Each hyperedge $e \in E$ corresponds to a subset of vertices, i.e., to an element of $2^V$. The vector $w \in \mathbb{R}^m$ contains for each hyperedge $e$ its non-negative weight $w_e$. In the following, we use the letter $H$ also for the incidence matrix $H \in \mathbb{R}^{|V| \times |E|}$ which is for $i \in V$ and $e \in E$, $H_{i,e} = \begin{cases} 1 & \text{if } i \in e, \\ 0 & \text{else.} \end{cases}$. The degree of a vertex $i \in V$ is defined as $d_i = \sum_{e \in E} w_e H_{i,e}$ and the cardinality of an edge $e$ can be written as $|e| = \sum_{j \in V} H_{j,e}$. We would like to emphasize that we do *not* impose the restriction that the hypergraph is $k$-uniform, i.e., that each hyperedge contains *exactly $k$* vertices.

The considered class of hypergraphs contains the set of undirected, weighted graphs which is equivalent to the set of 2-uniform hypergraphs. The motivation for the total variation on hypergraphs comes from the correspondence between the cut on a graph and the total variation functional. Thus, we recall the definition of the cut on weighted graphs $G = (V, W)$ with weight matrix $W$. Let $\overline{C} = V \backslash C$ denote the complement of $C$ in $V$. Then, for a partition $(C, \overline{C})$, the cut is defined as

$$\mathrm{cut}_G(C, \overline{C}) = \sum\nolimits_{i,j \,:\, i \in C, j \in \overline{C}} w_{ij}.$$

This standard definition of the cut carries over naturally to a hypergraph $H$

$$\mathrm{cut}_H(C, \overline{C}) = \sum_{\substack{e \in E: \\ e \cap C \neq \emptyset,\ e \cap \overline{C} \neq \emptyset}} w_e. \tag{1}$$

Thus, the cut functional on a hypergraph is just the sum of the weights of the hyperedges which have vertices both in $C$ and $\overline{C}$. It is not biased towards a particular way the hyperedge is cut, that is, how many vertices of the hyperedge are in $C$ resp. $\overline{C}$. This emphasizes that the vertices in a hyperedge belong together and we penalize every cut of a hyperedge with the same value.

In order to handle hypergraphs with existing methods developed for graphs, the focus in previous works [11, 12] has been on transforming the hypergraph into a graph. In [11], they suggest using the *clique expansion* (CE), i.e., every hyperedge $e \in H$ is replaced with a fully connected subgraph where every edge in this subgraph has weight $\frac{w_e}{|e|}$. This leads to the cut functional $\mathrm{cut}_{CE}$,

$$\mathrm{cut}_{CE}(C, \overline{C}) := \sum_{\substack{e \in E: \\ e \cap C \neq \emptyset,\ e \cap \overline{C} \neq \emptyset}} \frac{w_e}{|e|} |e \cap C| \, |e \cap \overline{C}|. \tag{2}$$

Note that in contrast to the hypergraph cut (1), the value of $\mathrm{cut}_{CE}$ depends on the way each hyperedge is cut since the term $|e \cap C| \, |e \cap \overline{C}|$ makes the weights dependent on the partition. In particular, the smallest weight is attained if only a single vertex is split off, whereas the largest weight is attained if the partition of the hyperedge is most balanced. In comparison to the hypergraph cut, this leads to a bias towards cuts that favor splitting off single vertices from a hyperedge which in our point of view is an undesired property for most applications. We illustrate this with an example in Figure 1, where the minimum hypergraph cut ($\mathrm{cut}_H$) leads to a balanced partition, whereas the minimum clique expansion cut ($\mathrm{cut}_{CE}$) not only cuts an additional hyperedge but is also unbalanced. This is due to its bias towards splitting off single nodes of a hyperedge. Another argument against the clique expansion is computational complexity. For large hyperedges the clique expansion leads to (almost) fully connected graphs which makes computations slow and is prohibitive for large hypergraphs.

We omit the discussion of the *star graph* approximation of hypergraphs discussed in [12] as it is shown there that the star graph expansion is very similar to the clique expansion. Instead, we want to recall the result of Ihler et al. [13] which states that in general there exists no graph with the same vertex set $V$ which has for every partition $(C, \overline{C})$ the same cut value as the hypergraph cut.

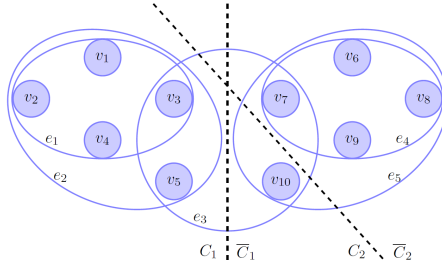

Figure 1: Minimum hypergraph cut $\mathrm{cut}_H$ vs. minimum cut of the clique expansion $\mathrm{cut}_{CE}$: For edge weights $w_1 = w_4 = 10$, $w_2 = w_5 = 0.1$ and $w_3 = 0.6$ the minimum hypergraph cut is $(C_1, \overline{C}_1)$ which is perfectly balanced. Although cutting one hyperedge more and being unbalanced, $(C_2, \overline{C}_2)$ is the optimal cut for the clique expansion approximation.

Finally, note that for weighted 3-uniform hypergraphs it is always possible to find a corresponding graph such that any cut of the graph is equal to the corresponding cut of the hypergraph.

**Proposition 2.1.** *Suppose $H = (V, E, w)$ is a weighted 3-uniform hypergraph. Then, $W \in \mathbb{R}^{|V| \times |V|}$ defined as $W = \frac{1}{2} H \mathrm{diag}(w) H^T$ defines the weight matrix of a graph $G = (V, W)$ where each cut of $G$ has the same value as the corresponding hypergraph cut of $H$.*

## 2.2 The Total Variation on Hypergraphs

In this section, we define the total variation on hypergraphs. The key technical element is the Lovasz extension which extends a set function, seen as a mapping on $2^V$, to a function on $\mathbb{R}^{|V|}$.

**Definition 2.1.** *Let $\hat{S} : 2^V \to \mathbb{R}$ be a set function with $\hat{S}(\emptyset) = 0$. Let $f \in \mathbb{R}^{|V|}$, let $V$ be ordered such that $f_1 \leq f_2 \leq \ldots \leq f_n$ and define $C_i = \{j \in V \mid j > i\}$. Then, the **Lovasz extension** $S : \mathbb{R}^{|V|} \to \mathbb{R}$ of $\hat{S}$ is given by*

$$S(f) = \sum_{i=1}^{n} f_i \Big( \hat{S}(C_{i-1}) - \hat{S}(C_i) \Big) = \sum_{i=1}^{n-1} \hat{S}(C_i)(f_{i+1} - f_i) + f_1 \hat{S}(V).$$

*Note that for the characteristic function of a set $C \subset V$, we have $S(\mathbf{1}_C) = \hat{S}(C)$.*

It is well-known that the Lovasz extension $S$ is a convex function if and only if $\hat{S}$ is submodular [18]. For graphs $G = (V, W)$, the total variation on graphs is defined as the Lovasz extension of the graph cut [18] given as $TV_G : \mathbb{R}^{|V|} \to \mathbb{R}$, $\mathrm{TV}_G(f) = \frac{1}{2} \sum_{i,j=1}^{n} w_{ij} |f_i - f_j|$.

**Proposition 2.2.** *The **total variation** $\mathrm{TV}_H : \mathbb{R}^{|V|} \to \mathbb{R}$ **on a hypergraph** $H = (V, E, w)$ defined as the Lovasz extension of the hypergraph cut, $\hat{S}(C) = \mathrm{cut}_H(C, \overline{C})$, is a convex function given by*

$$\mathrm{TV}_H(f) = \sum_{e \in E} w_e \Big( \max_{i \in e} f_i - \min_{j \in e} f_j \Big) = \sum_{e \in E} w_e \max_{i,j \in e} |f_i - f_j|.$$

Note that the total variation of a hypergraph cut reduces to the total variation on graphs if $H$ is 2-uniform (standard graph). There is an interesting relation of the total variation on hypergraphs to sparsity inducing group norms. Namely, defining for each edge $e \in E$ the difference operator $D_e : \mathbb{R}^{|V|} \to \mathbb{R}^{|V| \times |V|}$ by $(D_e f)_{ij} = f_i - f_j$ if $i, j \in e$ and 0 otherwise, $\mathrm{TV}_H$ can be written as, $\mathrm{TV}_H(f) = \sum_{e \in E} w_e \|D_e f\|_{\infty}$, which can be seen as inducing group sparse structure on the gradient level. The groups are the hyperedges and thus are typically overlapping. This could lead potentially to extensions of the elastic net on graphs to hypergraphs.

It is known that using the total variation on graphs as a regularization functional in semi-supervised learning (SSL) leads to very spiky solutions for small numbers of labeled points. Thus, one would like to have regularization functionals enforcing more smoothness of the solutions. For graphs this is achieved by using the family of regularization functionals $\Omega_{G,p} : \mathbb{R}^{|V|} \to \mathbb{R}$,

$$\Omega_{G,p}(f) = \frac{1}{2} \sum_{i,j=1}^{n} w_{ij} |f_i - f_j|^p.$$

For $p = 2$ we get the regularization functional of the graph Laplacian which is the basis of a large class of methods on graphs. In analogy to graphs, we define a corresponding family on hypergraphs.

**Definition 2.2.** *The regularization functionals* $\Omega_{H,p} : \mathbb{R}^{|V|} \to \mathbb{R}$ *for a hypergraph* $H = (V, E, w)$ *are defined for* $p \geq 1$ *as*

$$\Omega_{H,p}(f) = \sum_{e \in E} w_e \Big( \max_{i \in e} f_i - \min_{j \in e} f_j \Big)^p.$$

**Lemma 2.1.** *The functionals* $\Omega_{H,p} : \mathbb{R}^{|V|} \to \mathbb{R}$ *are convex.*

Note that $\Omega_{H,1}(f) = \mathrm{TV}_H(f)$. If $H$ is a graph and $p \geq 1$, $\Omega_{H,p}$ reduces to the Laplacian regularization $\Omega_{G,p}$. Note that for characteristic functions of sets, $f = \mathbf{1}_C$, it holds $\Omega_{H,p}(\mathbf{1}_C) = \mathrm{cut}_H(C, \overline{C})$. Thus, the difference between the hypergraph cut and its approximations such as clique and star expansion carries over to $\Omega_{H,p}$ and $\Omega_{G_{CE},p}$, respectively.

# 3 Semi-supervised Learning

With the regularization functionals derived in the last section, we can immediately write down a formulation for two-class semi-supervised learning on hypergraphs similar to the well-known approaches of [19, 20]. Given the label set $L$ we construct the vector $Y \in \mathbb{R}^n$ with $Y_i = 0$ if $i \notin L$ and $Y_i$ equal to the label in $\{-1, 1\}$ if $i \in L$. We propose solving

$$f^* = \underset{f \in \mathbb{R}^{|V|}}{\arg\min} \; \frac{1}{2} \|f - Y\|_2^2 + \lambda \, \Omega_{H,p}(f), \tag{3}$$

where $\lambda > 0$ is the regularization parameter. In Section 5, we discuss how this convex optimization problem can be solved efficiently for the case $p = 1$ and $p = 2$. Note, that other loss functions than the squared loss could be used. However, the regularizer aims at contracting the function and we use the label set $\{-1, 1\}$ so that $f^* \in [-1, 1]^{|V|}$. Hence, on the interval $[-1, 1]$ the squared loss behaves very similar to other margin-based loss functions. In general, we recommend using $p = 2$ as it corresponds to Laplacian-type regularization for graphs which is known to work well. For graphs $p = 1$ is known to produce spiky solutions for small numbers of labeled points. This is due to the effect that cutting "out" the labeled points leads to a much smaller cut than, e.g., producing a balanced partition. However, in the case where one has only a small number of hyperedges this effect is much smaller and we will see in the experiments that $p = 1$ also leads to reasonable solutions.

# 4 Balanced Hypergraph Cuts

In Section 2.1, we discussed the difference between the hypergraph cut (1) and the graph cut of the clique expansion (2) of the hypergraph and gave a simple example in Figure 1 where these cuts yield quite different results. Clearly, this difference carries over to the famous normalized cut criterion introduced in [21, 22] for clustering of graphs with applications in image segmentation. For a hypergraph the ratio resp. normalized cut can be formulated as

$$\mathrm{RCut}(C, \overline{C}) = \frac{\mathrm{cut}_H(C, \overline{C})}{|C||\overline{C}|}, \quad \mathrm{NCut}(C, \overline{C}) = \frac{\mathrm{cut}_H(C, \overline{C})}{\mathrm{vol}(C) \, \mathrm{vol}(\overline{C})},$$

which incorporate different balancing criteria. Note, that in contrast to the normalized cut for graphs the normalized hypergraph cut allows *no* relaxation into a linear eigenproblem (spectral relaxation).

Thus, we follow a recent line of research [14, 15, 16, 17] where it has been shown that the standard spectral relaxation of the normalized cut used in spectral clustering [22] is loose and that a tight, in fact exact, relaxation can be formulated in terms of a nonlinear eigenproblem. Although nonlinear eigenproblems are non-convex, one can compute nonlinear eigenvectors quite efficiently at the price of loosing global optimality. However, it has been shown that the potentially non-optimal solutions of the exact relaxation, outperform in practice the globally optimal solution of the loose relaxation, often by large margin. In this section, we extend their approach to hypergraphs and consider general balanced hypergraph cuts $\mathrm{Bcut}(C, \overline{C})$ of the form, $\mathrm{Bcut}(C, \overline{C}) = \frac{\mathrm{cut}_H(C, \overline{C})}{\hat{S}(C)}$, where $\hat{S} : 2^V \to \mathbb{R}_+$ is a non-negative, symmetric set function (that is $\hat{S}(C) = \hat{S}(\overline{C})$). For the normalized cut one has

$\hat{S}(C) = \mathrm{vol}(C)\,\mathrm{vol}(\overline{C})$ whereas for the Cheeger cut one has $\hat{S}(C) = \min\{\mathrm{vol}\,C, \mathrm{vol}\,\overline{C}\}$. Other examples of balancing functions can be found in [16]. Our following result shows that the balanced hypergraph cut also has an exact relaxation into a continuous nonlinear eigenproblem [14].

**Theorem 4.1.** *Let $H = (V, E, w)$ be a finite, weighted hypergraph and $S : \mathbb{R}^{|V|} \to \mathbb{R}$ be the Lovasz extension of the symmetric, non-negative set function $\hat{S} : 2^V \to \mathbb{R}$. Then, it holds that*

$$\min_{f \in \mathbb{R}^{|V|}} \frac{\sum_{e \in E} w_e\big(\max_{i \in e} f_i - \min_{j \in e} f_j\big)}{S(f)} = \min_{C \subset V} \frac{\mathrm{cut}_H(C, \overline{C})}{\hat{S}(C)}.$$

*Further, let $f \in \mathbb{R}^{|V|}$ and define $C_t := \{i \in V \mid f_i > t\}$. Then,*

$$\min_{t \in \mathbb{R}} \frac{\mathrm{cut}_H(C_t, \overline{C_t})}{\hat{S}(C_t)} \leq \frac{\sum_{e \in E} w_e\big(\max_{i \in e} f_i - \min_{j \in e} f_j\big)}{S(f)}.$$

The last part of the theorem shows that "optimal thresholding" (turning $f \in \mathbb{R}^V$ into a partition) among all level sets of any $f \in \mathbb{R}^{|V|}$ can only lead to a better or equal balanced hypergraph cut.

The question remains how to minimize the ratio $Q(f) = \frac{\mathrm{TV}_H(f)}{S(f)}$. As discussed in [16], every Lovasz extension $S$ can be written as a difference of convex positively 1-homogeneous functions[1] $S = S_1 - S_2$. Moreover, as shown in Prop. 2.2 the total variation $\mathrm{TV}_H$ is convex. Thus, we have to minimize a non-negative ratio of a convex and a difference of convex (d.c.) function. We employ the RatioDCA algorithm [16] shown in Algorithm 1. The main part is the convex inner problem. In

---

**Algorithm 1 RatioDCA** – Minimization of a non-negative ratio of 1-homogeneous d.c. functions

1: **Objective:** $Q(f) = \frac{R_1(f) - R_2(f)}{S_1(f) - S_2(f)}$. **Initialization:** $f^0 = $ random with $\|f^0\| = 1$, $\lambda^0 = Q(f^0)$
2: **repeat**
3: $\quad s_1(f^k) \in \partial S_1(f^k), r_2(f^k) \in \partial R_2(f^k)$
4: $\quad f^{k+1} = \underset{\|u\|_2 \leq 1}{\arg\min} \big\{ R_1(u) - \langle u, r_2(f^k) \rangle + \lambda^k \big( S_2(u) - \langle u, s_1(f^k) \rangle \big) \big\}$
5: $\quad \lambda^{k+1} = (R_1(f^{k+1}) - R_2(f^{k+1}))/(S_1(f^{k+1}) - S_2(f^{k+1}))$
6: **until** $\frac{|\lambda^{k+1} - \lambda^k|}{\lambda^k} < \epsilon$
7: **Output:** eigenvalue $\lambda^{k+1}$ and eigenvector $f^{k+1}$.

---

our case $R_1 = TV_H, R_2 = 0$, and thus the inner problem reads

$$\min_{\|u\|_2 \leq 1}\{\mathrm{TV}_H(u) + \lambda^k \big( S_2(u) - \langle u, s_1(f^k) \rangle \big)\}. \tag{4}$$

For simplicity we restrict ourselves to submodular balancing functions, in which case $S$ is convex and thus $S_2 = 0$. For the general case, see [16]. Note that the balancing functions of ratio/normalized cut and Cheeger cut are submodular. It turns out that the inner problem is very similar to the semi-supervised learning formulation (3). The efficient solution of both problems is discussed next.

## 5 Algorithms for the Total Variation on Hypergraphs

The problem (3) we want to solve for semi-supervised learning and the inner problem (4) of RatioDCA have a common structure. They are the sum of two convex functions: one of them is the novel regularizer $\Omega_{H,p}$ and the other is a data term denoted by $G$ here, cf., Table 1. We propose solving these problems using a primal-dual algorithm, denoted by PDHG, which was proposed in [23, 24]. Its main idea is to iteratively solve for each convex term in the objective function a proximal problem. The proximal map $\mathrm{prox}_g$ w.r.t. a mapping $g : \mathbb{R}^n \to \mathbb{R}$ is defined by

$$\mathrm{prox}_g(\tilde{x}) = \underset{x \in \mathbb{R}^n}{\arg\min}\{\frac{1}{2}\|x - \tilde{x}\|_2^2 + g(x)\}.$$

The key idea is that often proximal problems can be solved efficiently leading to fast convergence of the overall algorithm. We see in Table 1 that for both $G$ the proximal problems have an explicit solution. However, note that smooth convex terms can also be directly exploited [25]. For $\Omega_{H,p}$, we distinguish two cases, $p = 1$ and $p = 2$. Detailed descriptions of the algorithms can be found in the supplementary material.

| $G(f) = \frac{1}{2}\|f - Y\|_2^2$ | $G(f) = -\langle s_1(f^k), f\rangle + \iota_{\|\cdot\|_2 \leq 1}(f)$ |
|---|---|
| $\mathrm{prox}_{\tau G(f)}(\tilde{x}) = \frac{1}{1+\tau}(\tilde{x} + \tau Y)$ | $\mathrm{prox}_{\tau G(f)}(\tilde{x}) = \frac{\tilde{x} + \tau s_1(f^k)}{\max\{1, \|\tilde{x} + \tau s_1(f^k)\|_2\}}$ |

Table 1: Data term and proximal map for SSL (3) (left) and the inner problem of RatioDCA (4) (right). The indicator function is defined as $\iota_{\|\cdot\|_2 \leq 1}(x) = 0$, if $\|x\|_2 \leq 1$ and $+\infty$ otherwise.

**PDHG algorithm for $\Omega_{H,1}$.** Let $m_e$ be the number of vertices in hyperedge $e \in E$. We write

$$\lambda\Omega_{H,1}(f) = F(Kf) := \sum_{e \in E}(F_{(e,1)}(K_e f) + F_{(e,2)}(K_e f)), \tag{5}$$

where the rows of the matrices $K_e \in \mathbb{R}^{m_e, n}$ are the $i$-th standard unit vectors for $i \in e$ and the functionals $F_{(e,j)} : \mathbb{R}^{m_e} \to \mathbb{R}$ are defined as

$$F_{(e,1)}(\alpha^{(e,1)}) = \lambda w_e \max(\alpha^{(e,1)}), \quad F_{(e,2)}(\alpha^{(e,2)}) = -\lambda w_e \min(\alpha^{(e,2)}).$$

In contrast to the function $G$, we need in the PDHG algorithm the proximal maps for the conjugate functions of $F_{(e,j)}$. They are given by

$$F_{(e,1)}^* = \iota_{S_{\lambda w_e}}, \quad F_{(e,2)}^* = \iota_{-S_{\lambda w_e}},$$

where $S_{\lambda w_e} = \{x \in \mathbb{R}^{m_e} : \sum_{i=1}^{m_e} x_i = \lambda w_e, x_i \geq 0\}$ is the scaled simplex in $\mathbb{R}^{m_e}$. The solutions of the proximal problem for $F_{(e,1)}^*$ and $F_{(e,1)}^*$ are the orthogonal projections onto the simplexes written here as $P_{S_{\lambda w_e}^e}$ and $P_{-S_{\lambda w_e}^e}$, respectively. These projections can be done in linear time [26]. With the proximal maps we have presented so far, the PDHG algorithm has the following form.

---

**Algorithm 2 PDHG for $\Omega_{H,1}$**

1: **Initialization:** $f^{(0)} = \bar{f}^{(0)} = 0$, $\theta \in [0,1]$, $\sigma, \tau > 0$ with $\sigma\tau < 1/(2\max_{i=1,\ldots,n}\{c_i\})$
2: **repeat**
3: $\quad \alpha^{(e,1)(k+1)} = P_{S_{\lambda w_e}^e}(\alpha^{(e,1)(k)} + \sigma K_e \bar{f}^{(k)}), \quad e \in E$
4: $\quad \alpha^{(e,2)(k+1)} = P_{-S_{\lambda w_e}^e}(\alpha^{(e,2)(k)} + \sigma K_e \bar{f}^{(k)}), \quad e \in E$
5: $\quad f^{(k+1)} = \mathrm{prox}_{\tau G}(f^{(k)} - \tau \sum_{e \in E} K_e^{\mathrm{T}}(\alpha^{(e,1)(k+1)} + \alpha^{(e,2)(k+1)}))$
6: $\quad \bar{f}^{(k+1)} = f^{(k+1)} + \theta(f^{(k+1)} - f^{(k)})$
7: **until** relative duality gap $< \epsilon$
8: **Output:** $f^{(k+1)}$.

---

The value $c_i = \sum_{e \in E} H_{i,e}$ is the number of hyperedges the vertex $i$ lies in. It is important to point out here that the algorithm decouples the problem in the sense that in every iteration we solve subproblems which treat the functionals $G, F_{(e,1)}, F_{(e,2)}$ separately and thus can be solved efficiently.

**PDHG algorithm for $\Omega_{H,2}$.** We define the matrices $K_e$ as above. Moreover, we introduce for every hyperedge $e \in E$ the functional

$$F_e(\alpha^e) = \lambda w_e(\max(\alpha^e) - \min(\alpha^e))^2. \tag{6}$$

Hence, we can write $\Omega_{H,2}(f) = \sum_{e \in E} F_e(K_e f)$. As we show in the supplementary material, the conjugate functions $F_e^*$ are not indicator functions and we thus solve the corresponding proximal problems via proximal problems for $F_e$. More specifically, we exploit the fact that

$$\mathrm{prox}_{\sigma F_e^*}(\tilde{\alpha}^e) = \tilde{\alpha}^e - \mathrm{prox}_{\frac{1}{\sigma}F_e}(\tilde{\alpha}^e), \tag{7}$$

and use the following novel result concerning the proximal problem on the right-hand side of (7).

| Prop. \ Dataset | Zoo | Mushrooms | Covertype (4,5) | Covertype (6,7) | 20Newsgroups |
|---|---|---|---|---|---|
| Number of classes | 7 | 2 | 2 | 2 | 4 |
| $|V|$ | 101 | 8124 | 12240 | 37877 | 16242 |
| $|E|$ | 42 | 112 | 104 | 123 | 100 |
| $\sum_{e \in E} |e|$ | 1717 | 170604 | 146880 | 454522 | 65451 |
| $|E|$ of Clique Exp. | 10201 | 65999376 | 143008092 | 1348219153 | 53284642 |

Table 2: Datasets used for SSL and clustering. Note that the clique expansion leads for all datasets to a graph which is close to being fully connected as all datasets contain large hyperedges. For covertype (6,7) the weight matrix needs over 10GB of memory, the original hypergraph only 4MB.

**Proposition 5.1.** *For any $\sigma > 0$ and any $\tilde{\alpha}^e \in \mathbb{R}^{m_e}$ the proximal map*

$$\mathrm{prox}_{\frac{1}{\sigma}F_e}(\tilde{\alpha}^e) = \arg\min_{\alpha^e \in \mathbb{R}^{m_e}} \{\frac{1}{2}\|\alpha^e - \tilde{\alpha}^e\|_2^2 + \frac{1}{\sigma}\lambda w_e(\max(\alpha^e) - \min(\alpha^e))^2\}$$

*can be computed with $\mathcal{O}(m_e \log m_e)$ arithmetic operations.*

A corresponding algorithm which is new to the best of our knowledge is provided in the supplementary material. We note here that the complexity is due to the fact that we sort the input vector $\tilde{\alpha}^e$. The PDHG algorithm for $p = 2$ is provided in the supplementary material. It has the same structure as Algorithm 2 with the only difference that we now solve (7) for every hyperedge.

## 6 Experiments

The method of Zhou et al [11] seems to be the standard algorithm for clustering and SSL on hypergraphs. We compare to them on a selection of UCI datasets summarized in Table 2. Zoo, Mushrooms and 20Newsgroups[2] have been used also in [11] and contain only categorical features. As in [11], a hyperedge of weight one is created by all data points which have the same value of a categorical feature. For covertype we quantize the numerical features into 10 bins of equal size. Two datasets are created each with two classes (4,5 and 6,7) of the original dataset.

**Semi-supervised Learning (SSL).**   In [11], they suggest using a regularizer induced by the normalized Laplacian $L_{CE}$ arising from the clique expansion

$$L_{CE} = \mathbb{I} - D_{CE}^{-\frac{1}{2}}HW'H^T D_{CE}^{-\frac{1}{2}},$$

where $D_{CE}$ is a diagonal matrix with entries $d_{EC}(i) = \sum_{e \in E} H_{i,e}\frac{w_e}{|e|}$ and $W' \in \mathbb{R}^{|E| \times |E|}$ is a diagonal matrix with entries $w'(e) = w_e/|e|$. The SSL problem can then be formulated as

$$\lambda > 0, \qquad \arg\min_{f \in \mathbb{R}^{|V|}} \{\|f - Y\|_2^2 + \lambda \langle f, L_{CE}f \rangle\}.$$

The advantage of this formulation is that the solution can be found via a linear system. However, as Table 2 indicates the obvious downside is that $L_{CE}$ is a potentially very dense matrix and thus one needs in the worst case $|V|^2$ memory and $O(|V|^3)$ computations. This is in contrast to our method which needs $2\sum_{e \in E} |e| + |V|$ memory. For the largest example (covertype 6,7), where the clique expansion fails due to memory problems, our method takes 30-100s (depending on $\lambda$). We stop our method for all experiments when we achieve a relative duality gap of $10^{-6}$. In the experiments we do 10 trials for different numbers of labeled points. The reg. parameter $\lambda$ is chosen for both methods from the set $10^{-k}$, where $k = \{0, 1, 2, 3, 4, 5, 6\}$ via 5-fold cross validation. The resulting errors and standard deviations can be found in the following table(first row lists the no. of labeled points).

Our SSL methods based on $\Omega_{H,p}$, $p = 1, 2$ outperform consistently the clique expansion technique of Zhou et al [11] on all datasets except 20newsgroups[3]. However, 20newsgroups is a very difficult dataset as only 10,267 out of the 16,242 data points are different which leads to a minimum possible error of 9.6%. A method based on pairwise interaction such as the clique expansion can better deal

with such label noise as the large hyperedges for this dataset accumulate the label noise. On all other datasets we observe that incorporating hypergraph structure leads to much better results. As expected our squared TV functional ($p = 2$) outperforms slightly the total variation ($p = 1$) even though the difference is small. Thus, as $\Omega_{H,2}$ reduces to the standard regularization based on the graph Laplacian, which is known to work well, we recommend $\Omega_{H,2}$ for SSL on hypergraphs.

| Zoo | 20 | 25 | 30 | 35 | 40 | 45 | 50 |
|---|---|---|---|---|---|---|---|
| Zhou et al. | $35.1 \pm 17.2$ | $30.3 \pm 7.9$ | $40.7 \pm 14.2$ | $29.7 \pm 8.8$ | $32.9 \pm 16.8$ | $27.6 \pm 10.8$ | $25.3 \pm 14.4$ |
| $\Omega_{H,1}$ | $2.9 \pm 3.0$ | $\mathbf{1.4 \pm 2.2}$ | $\mathbf{2.2 \pm 2.1}$ | $\mathbf{0.7 \pm 1.0}$ | $\mathbf{0.7 \pm 1.5}$ | $\mathbf{0.9 \pm 1.4}$ | $1.9 \pm 3.0$ |
| $\Omega_{H,2}$ | $\mathbf{2.3 \pm 1.9}$ | $1.5 \pm 2.4$ | $2.9 \pm 2.3$ | $0.9 \pm 1.4$ | $0.8 \pm 1.7$ | $1.2 \pm 1.8$ | $\mathbf{1.6 \pm 2.9}$ |

| Mushr. | 20 | 40 | 60 | 80 | 100 | 120 | 160 | 200 |
|---|---|---|---|---|---|---|---|---|
| Zhou et al. | $\mathbf{15.5 \pm 12.8}$ | $10.9 \pm 4.4$ | $9.5 \pm 2.7$ | $10.3 \pm 2.0$ | $9.0 \pm 4.5$ | $8.8 \pm 1.4$ | $8.8 \pm 2.3$ | $9.3 \pm 1.0$ |
| $\Omega_{H,1}$ | $19.5 \pm 10.5$ | $10.8 \pm 3.7$ | $\mathbf{7.4 \pm 3.8}$ | $\mathbf{5.6 \pm 1.9}$ | $\mathbf{5.7 \pm 2.2}$ | $5.4 \pm 2.4$ | $4.9 \pm 3.8$ | $5.6 \pm 3.8$ |
| $\Omega_{H,2}$ | $18.4 \pm 7.4$ | $\mathbf{9.8 \pm 4.5}$ | $9.9 \pm 5.5$ | $6.4 \pm 2.7$ | $6.3 \pm 2.5$ | $\mathbf{4.5 \pm 1.8}$ | $\mathbf{4.4 \pm 2.1}$ | $\mathbf{3.0 \pm 0.6}$ |
| covert45 | 20 | 40 | 60 | 80 | 100 | 120 | 160 | 200 |
| Zhou et al. | $\mathbf{18.9 \pm 4.6}$ | $18.3 \pm 5.2$ | $17.2 \pm 6.7$ | $16.6 \pm 6.4$ | $17.6 \pm 5.2$ | $18.4 \pm 5.1$ | $19.2 \pm 4.0$ | $20.4 \pm 2.9$ |
| $\Omega_{H,1}$ | $21.4 \pm 0.9$ | $17.6 \pm 2.6$ | $12.6 \pm 4.3$ | $7.6 \pm 3.5$ | $6.2 \pm 3.8$ | $4.5 \pm 3.6$ | $2.6 \pm 1.6$ | $1.5 \pm 1.3$ |
| $\Omega_{H,2}$ | $20.7 \pm 2.0$ | $\mathbf{16.1 \pm 4.1}$ | $\mathbf{10.9 \pm 4.9}$ | $\mathbf{5.9 \pm 3.7}$ | $\mathbf{4.6 \pm 3.4}$ | $\mathbf{3.3 \pm 3.1}$ | $\mathbf{2.2 \pm 1.8}$ | $\mathbf{1.0 \pm 1.1}$ |
| covert67 | 20 | 40 | 60 | 80 | 100 | 120 | 160 | 200 |
| $\Omega_{H,1}$ | $40.6 \pm 8.9$ | $6.4 \pm 10.4$ | $3.6 \pm 3.2$ | $3.3 \pm 2.5$ | $1.8 \pm 0.8$ | $1.3 \pm 0.9$ | $0.9 \pm 0.4$ | $1.2 \pm 0.9$ |
| $\Omega_{H,2}$ | $\mathbf{25.2 \pm 18.3}$ | $\mathbf{4.3 \pm 9.6}$ | $\mathbf{2.1 \pm 2.0}$ | $\mathbf{2.2 \pm 1.4}$ | $\mathbf{1.4 \pm 1.1}$ | $\mathbf{1.0 \pm 0.8}$ | $\mathbf{0.7 \pm 0.4}$ | $\mathbf{1.1 \pm 0.8}$ |
| 20news | 20 | 40 | 60 | 80 | 100 | 120 | 160 | 200 |
| Zhou et al. | $\mathbf{45.5 \pm 7.5}$ | $\mathbf{34.4 \pm 3.1}$ | $\mathbf{31.5 \pm 1.4}$ | $\mathbf{29.8 \pm 4.0}$ | $\mathbf{27.0 \pm 1.3}$ | $\mathbf{27.3 \pm 1.5}$ | $\mathbf{25.7 \pm 1.4}$ | $\mathbf{25.0 \pm 1.3}$ |
| $\Omega_{H,1}$ | $65.7 \pm 6.1$ | $61.4 \pm 6.1$ | $53.2 \pm 5.7$ | $46.2 \pm 3.7$ | $42.4 \pm 3.3$ | $40.9 \pm 3.2$ | $36.1 \pm 1.5$ | $34.7 \pm 3.6$ |
| $\Omega_{H,2}$ | $55.0 \pm 4.8$ | $48.0 \pm 6.0$ | $45.0 \pm 5.9$ | $40.3 \pm 3.0$ | $38.3 \pm 2.7$ | $38.1 \pm 2.6$ | $35.0 \pm 2.8$ | $34.1 \pm 2.0$ |

Test error and standard deviation of the SSL methods over 10 runs for varying number of labeled points.

**Clustering.** We use the normalized hypergraph cut as clustering objective. For more than two clusters we recursively partition the hypergraph until the desired number of clusters is reached. For comparison we use the normalized spectral clustering approach based on the Laplacian $L_{CE}$ [11](clique expansion). The first part (first 6 columns) of the following table shows the clustering errors (majority vote on each cluster) of both methods as well as the normalized cuts achieved by these methods on the hypergraph and on the graph resulting from the clique expansion. Moreover, we show results (last 4 columns) which are obtained based on a $k$NN graph (unit weights) which is built based on the Hamming distance (note that we have categorical features) in order to check if the hypergraph modeling of the problem is actually useful compared to a standard similarity based graph construction. The number $k$ is chosen as the smallest number for which the graph becomes connected and we compare results of normalized 1-spectral clustering [14] and the standard spectral clustering [22]. Note that the employed hypergraph construction has no free parameter.

| Dataset | Clustering Error % | | Hypergraph Ncut | | Graph(CE) Ncut | | Clustering Error % | | kNN-Graph Ncut | |
|---|---|---|---|---|---|---|---|---|---|---|
|  | Ours | [11] | Ours | [11] | Ours | [11] | [14] | [22] | [14] | [22] |
| Mushrooms | 10.98 | 32.25 | 0.0011 | 0.0013 | 0.6991 | 0.7053 | 48.2 | 48.2 | 1e-4 | 1e-4 |
| Zoo | 16.83 | 15.84 | 0.6739 | 0.6784 | 5.1315 | 5.1703 | 5.94 | 5.94 | 1.636 | 1.636 |
| 20-newsgroup | 47.77 | 33.20 | 0.0176 | 0.0303 | 2.3846 | 1.8492 | 66.38 | 66.38 | 0.1031 | 0.1034 |
| covertype (4,5) | 22.44 | 22.44 | 0.0018 | 0.0022 | 0.7400 | 0.6691 | 22.44 | 22.44 | 0.0152 | 0.02182 |
| covertype (6,7) | 8.16 | - | 8.18e-4 | - | 0.6882 | - | 45.85 | 45.85 | 0.0041 | 0.0041 |

First, we observe that our approach optimizing the normalized hypergraph cut yields better or similar results in terms of clustering errors compared to the clique expansion (except for 20-newsgroup for the same reason given in the previous paragraph). The improvement is significant in case of Mushrooms while for Zoo our clustering error is slightly higher. However, we always achieve smaller normalized hypergraph cuts. Moreover, our method sometimes has even smaller cuts on the graphs resulting from the clique expansion, although it does not directly optimize this objective in contrast to [11]. Again, we could not run the method of [11] on covertype (6,7) since the weight matrix is very dense. Second, the comparison to a standard graph-based approach where the similarity structure is obtained using the Hamming distance on the categorical features shows that using hypergraph structure is indeed useful. Nevertheless, we think that there is room for improvement regarding the construction of the hypergraph which is a topic for future research.

**Acknowledgments**

M.H. would like to acknowledge support by the ERC Starting Grant NOLEPRO and L.J. acknowledges support by the DFG SPP-1324.

## Footnotes

[1]A function $f : \mathbb{R}^d \to \mathbb{R}$ is positively 1-homogeneous if $\forall \alpha > 0, f(\alpha x) = \alpha f(x)$.

[2]This is a modified version by Sam Roweis of the original 20 newsgroups dataset available at http://www.cs.nyu.edu/~roweis/data/20news_w100.mat.

[3]Communications with the authors of [11] could not clarify the difference to their results on 20newsgroups

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
