[Supplementary Material]

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

*Proof.* The cut value of a partition $(C, \overline{C})$ of $G$ is given as

$$\mathrm{cut}_G(C, \overline{C}) = \frac{1}{2} \sum_{e \in E} |e \cap C||e \cap \overline{C}|w_e.$$

The product $|e \cap C||e \cap \overline{C}|$ takes the values 2 if $e$ is cut by $C$ and zero otherwise. Because of the factor $\frac{1}{2}$, we thus get equivalence to the hypergraph cut. □

## 2.2 The Total Variation on Hypergraphs

In this section, we define the total variation on hypergraphs. The key technical element is the Lovasz extension which extends a set function, seen as a mapping on $2^V$, to a function on $\mathbb{R}^{|V|}$.

**Definition 2.1.** *Let $\hat{S} : 2^V \to \mathbb{R}$ be a set function with $\hat{S}(\emptyset) = 0$. Let $f \in \mathbb{R}^{|V|}$, let $V$ be ordered such that $f_1 \leq f_2 \leq \ldots \leq f_n$ and define $C_i = \{j \in V \mid j > i\}$. Then, the **Lovasz extension** $S : \mathbb{R}^{|V|} \to \mathbb{R}$ of $\hat{S}$ is given by*

$$S(f) = \sum_{i=1}^{n} f_i \Big( \hat{S}(C_{i-1}) - \hat{S}(C_i) \Big) = \sum_{i=1}^{n-1} \hat{S}(C_i)(f_{i+1} - f_i) + f_1 \hat{S}(V).$$

*Note that for the characteristic function of a set $C \subset V$, we have $S(\mathbf{1}_C) = \hat{S}(C)$.*

It is well-known that the Lovasz extension $S$ is a convex function if and only if $\hat{S}$ is submodular [18]. For graphs $G = (V, W)$, the total variation on graphs is defined as the Lovasz extension of the graph cut [18] given as $TV_G : \mathbb{R}^{|V|} \to \mathbb{R}$, $TV_G(f) = \frac{1}{2} \sum_{i,j=1}^{n} w_{ij} |f_i - f_j|$.

**Proposition 2.2.** *The **total variation** $\mathrm{TV}_H : \mathbb{R}^{|V|} \to \mathbb{R}$ **on a hypergraph** $H = (V, E, w)$ defined as the Lovasz extension of the hypergraph cut, $\hat{S}(C) = \mathrm{cut}_H(C, \overline{C})$, is a convex function given by*

$$\mathrm{TV}_H(f) = \sum_{e \in E} w_e \Big( \max_{i \in e} f_i - \min_{j \in e} f_j \Big) = \sum_{e \in E} w_e \max_{i,j \in e} |f_i - f_j|.$$

*Proof.* Using $C_{i-1} = C_i \cup \{i\}$ and $\overline{C_i} = \overline{C_{i-1}} \cup \{i\}$ the Lovasz extension can be written as

$$\mathrm{TV}_H(f) = \sum_{i=1}^{n} f_i \Big( \mathrm{cut}(C_{i-1}, \overline{C_{i-1}}) - \mathrm{cut}(C_i, \overline{C_i}) \Big) = \sum_{i=1}^{n} f_i \Big( \mathrm{cut}(\{i\}, \overline{C_{i-1}}) - \mathrm{cut}(C_i, \{i\}) \Big)$$

$$= \sum_{i=1}^{n} f_i \Big( \sum_{\substack{e \in E, i \in e \\ e \cap \{1, \ldots, i-1\} \neq \emptyset}} w_e - \sum_{\substack{e \in E, i \in e \\ e \cap \{i+1, \ldots, n\} \neq \emptyset}} w_e \Big) = \sum_{e \in E} w_e \Big( \max_{i \in e} f_i - \min_{j \in e} f_j \Big).$$

It is easy to see that the Lovasz extension of the hypergraph cut is a convex function. Since the maximum of convex functions is convex, $-\min_{i \in e} f_i = \max_{i \in e} f_i$ and the hyperedge weights are non-negative, we have a non-negative combination of convex functions which is convex. Alternatively, one could use that the hypergraph cut is submodular and the Lovasz extension of every submodular set function is convex. $\qquad\square$

Note that the total variation of a hypergraph cut reduces to the total variation on graphs if $H$ is 2-uniform (standard graph). There is an interesting relation of the total variation on hypergraphs to sparsity inducing group norms. Namely, defining for each edge $e \in E$ the difference operator $D_e : \mathbb{R}^{|V|} \to \mathbb{R}^{|V| \times |V|}$ by $(D_e f)_{ij} = f_i - f_j$ if $i, j \in e$ and 0 otherwise, $\mathrm{TV}_H$ can be written as, $\mathrm{TV}_H(f) = \sum_{e \in E} w_e \|D_e f\|_\infty$, which can be seen as inducing group sparse structure on the gradient level. The groups are the hyperedges and thus are typically overlapping. This could lead potentially to extensions of the elastic net on graphs to hypergraphs.

It is known that using the total variation on graphs as a regularization functional in semi-supervised learning (SSL) leads to very spiky solutions for small numbers of labeled points. Thus, one would like to have regularization functionals enforcing more smoothness of the solutions. For graphs this is achieved by using the family of regularization functionals $\Omega_{G,p} : \mathbb{R}^{|V|} \to \mathbb{R}$,

$$\Omega_{G,p}(f) = \frac{1}{2} \sum_{i,j=1}^{n} w_{ij} |f_i - f_j|^p.$$

For $p = 2$ we get the regularization functional of the graph Laplacian which is the basis of a large class of methods on graphs. In analogy to graphs, we define a corresponding family on hypergraphs.

**Definition 2.2.** *The regularization functionals* $\Omega_{H,p} : \mathbb{R}^{|V|} \to \mathbb{R}$ *for a hypergraph* $H = (V, E, w)$ *are defined for* $p \geq 1$ *as*

$$\Omega_{H,p}(f) = \sum_{e \in E} w_e \left( \max_{i \in e} f_i - \min_{j \in e} f_j \right)^p.$$

**Lemma 2.1.** *The functionals* $\Omega_{H,p} : \mathbb{R}^{|V|} \to \mathbb{R}$ *are convex.*

*Proof.* The $p$-th power of positive, convex functions for $p \geq 1$ is convex as

$$\left( f(\lambda x + (1 - \lambda) y) \right)^p \leq \left( \lambda f(x) + (1 - \lambda) f(y) \right)^p \leq \lambda f(x)^p + (1 - \lambda) f(y)^p$$

where the last inequality follows from the convexity of $x^p$ on $\mathbb{R}_+$. Thus, the $p$-th power of $\max_{i \in e} f_i - \min_{j \in e} f_j$ is convex. $\qquad\square$

Note that $\Omega_{H,1}(f) = \mathrm{TV}_H(f)$. If $H$ is a graph and $p \geq 1$, $\Omega_{H,p}$ reduces to the Laplacian regularization $\Omega_{G,p}$. Note that for characteristic functions of sets, $f = \mathbf{1}_C$, it holds $\Omega_{H,p}(\mathbf{1}_C) = \mathrm{cut}_H(C, \overline{C})$. Thus, the difference between the hypergraph cut and its approximations such as clique and star expansion carries over to $\Omega_{H,p}$ and $\Omega_{G_{CE},p}$, respectively.

## 3 Semi-supervised Learning

With the regularization functionals derived in the last section, we can immediately write down a formulation for two-class semi-supervised learning on hypergraphs similar to the well-known approaches of [19, 20]. Given the label set $L$ we construct the vector $Y \in \mathbb{R}^n$ with $Y_i = 0$ if $i \notin L$ and $Y_i$ equal to the label in $\{-1, 1\}$ if $i \in L$. We propose solving

$$f^* = \underset{f \in \mathbb{R}^{|V|}}{\arg\min} \, \frac{1}{2} \|f - Y\|_2^2 + \lambda \, \Omega_{H,p}(f), \tag{3}$$

where $\lambda > 0$ is the regularization parameter. In Section 5, we discuss how this convex optimization problem can be solved efficiently for the case $p = 1$ and $p = 2$. Note, that other loss functions than the squared loss could be used. However, the regularizer aims at contracting the function and we use the label set $\{-1, 1\}$ so that $f^* \in [-1, 1]^{|V|}$. Hence, on the interval $[-1, 1]$ the squared loss behaves very similar to other margin-based loss functions. In general, we recommend using $p = 2$

as it corresponds to Laplacian-type regularization for graphs which is known to work well. For graphs $p = 1$ is known to produce spiky solutions for small numbers of labeled points. This is due to the effect that cutting "out" the labeled points leads to a much smaller cut than, e.g., producing a balanced partition. However, in the case where one has only a small number of hyperedges this effect is much smaller and we will see in the experiments that $p = 1$ also leads to reasonable solutions.

## 4 Balanced Hypergraph Cuts

In Section 2.1, we discussed the difference between the hypergraph cut (1) and the graph cut of the clique expansion (2) of the hypergraph and gave a simple example in Figure 1 where these cuts yield quite different results. Clearly, this difference carries over to the famous normalized cut criterion introduced in [21, 22] for clustering of graphs with applications in image segmentation. For a hypergraph the ratio resp. normalized cut can be formulated as

$$\text{RCut}(C, \overline{C}) = \frac{\text{cut}_H(C, \overline{C})}{|C||\overline{C}|}, \quad \text{NCut}(C, \overline{C}) = \frac{\text{cut}_H(C, \overline{C})}{\text{vol}(C)\,\text{vol}(\overline{C})},$$

which incorporate different balancing criteria. Note, that in contrast to the normalized cut for graphs the normalized hypergraph cut allows *no* relaxation into a linear eigenproblem (spectral relaxation).

Thus, we follow a recent line of research [14, 15, 16, 17] where it has been shown that the standard spectral relaxation of the normalized cut used in spectral clustering [22] is loose and that a tight, in fact exact, relaxation can be formulated in terms of a nonlinear eigenproblem. Although nonlinear eigenproblems are non-convex, one can compute nonlinear eigenvectors quite efficiently at the price of loosing global optimality. However, it has been shown that the potentially non-optimal solutions of the exact relaxation, outperform in practice the globally optimal solution of the loose relaxation, often by large margin. In this section, we extend their approach to hypergraphs and consider general balanced hypergraph cuts $\text{Bcut}(C, \overline{C})$ of the form, $\text{Bcut}(C, \overline{C}) = \frac{\text{cut}_H(C,\overline{C})}{\hat{S}(C)}$, where $\hat{S} : 2^V \to \mathbb{R}_+$ is a non-negative, symmetric set function (that is $\hat{S}(C) = \hat{S}(\overline{C})$). For the normalized cut one has $\hat{S}(C) = \text{vol}(C)\,\text{vol}(\overline{C})$ whereas for the Cheeger cut one has $\hat{S}(C) = \min\{\text{vol}\,C, \text{vol}\,\overline{C}\}$. Other examples of balancing functions can be found in [16]. Our following result shows that the balanced hypergraph cut also has an exact relaxation into a continuous nonlinear eigenproblem [14].

**Theorem 4.1.** *Let $H = (V, E, w)$ be a finite, weighted hypergraph and $S : \mathbb{R}^{|V|} \to \mathbb{R}$ be the Lovasz extension of the symmetric, non-negative set function $\hat{S} : 2^V \to \mathbb{R}$. Then, it holds that*

$$\min_{f \in \mathbb{R}^{|V|}} \frac{\sum_{e \in E} w_e \big( \max_{i \in e} f_i - \min_{j \in e} f_j \big)}{S(f)} = \min_{C \subset V} \frac{\text{cut}_H(C, \overline{C})}{\hat{S}(C)}.$$

*Further, let $f \in \mathbb{R}^{|V|}$ and define $C_t := \{i \in V \mid f_i > t\}$. Then,*

$$\min_{t \in \mathbb{R}} \frac{\text{cut}_H(C_t, \overline{C_t})}{\hat{S}(C_t)} \leq \frac{\sum_{e \in E} w_e \big( \max_{i \in e} f_i - \min_{j \in e} f_j \big)}{S(f)}.$$

*Proof.* By Prop. 2.2 the Lovasz extension of $\text{cut}_H(C, \overline{C})$ is given by $\sum_{e \in E} w_e \big( \max_{i \in e} f_i - \min_{j \in e} f_j \big)$.

Noting that both $\text{cut}_H(C, \overline{C})$ and $\hat{S}(C)$ vanish on the full set $V$, the proof then follows from the recent result [17], which shows in this case the equivalence between the set problem and the continuous problem written in terms of the Lovasz extensions. $\square$

The last part of the theorem shows that "optimal thresholding" (turning $f \in \mathbb{R}^V$ into a partition) among all level sets of any $f \in \mathbb{R}^{|V|}$ can only lead to a better or equal balanced hypergraph cut.

The question remains how to minimize the ratio $Q(f) = \frac{\text{TV}_H(f)}{S(f)}$. As discussed in [16], every Lovasz extension $S$ can be written as a difference of convex positively 1-homogeneous functions[1] $S = S_1 - S_2$. Moreover, as shown in Prop. 2.2 the total variation $\text{TV}_H$ is convex. Thus, we have to minimize a non-negative ratio of a convex and a difference of convex (d.c.) function. We employ

---

**Algorithm 1 RatioDCA** – Minimization of a non-negative ratio of 1-homogeneous d.c. functions

---

1: **Objective:** $Q(f) = \frac{R_1(f) - R_2(f)}{S_1(f) - S_2(f)}$. **Initialization:** $f^0 =$ random with $\|f^0\| = 1$, $\lambda^0 = Q(f^0)$

2: **repeat**

3:     $s_1(f^k) \in \partial S_1(f^k)$, $r_2(f^k) \in \partial R_2(f^k)$

4:     $f^{k+1} = \underset{\|u\|_2 \leq 1}{\arg\min} \left\{ R_1(u) - \langle u, r_2(f^k) \rangle + \lambda^k \left( S_2(u) - \langle u, s_1(f^k) \rangle \right) \right\}$

5:     $\lambda^{k+1} = (R_1(f^{k+1}) - R_2(f^{k+1}))/(S_1(f^{k+1}) - S_2(f^{k+1}))$

6: **until** $\frac{|\lambda^{k+1} - \lambda^k|}{\lambda^k} < \epsilon$

7: **Output:** eigenvalue $\lambda^{k+1}$ and eigenvector $f^{k+1}$.

---

the RatioDCA algorithm [16] shown in Algorithm 1. The main part is the convex inner problem. In our case $R_1 = TV_H$, $R_2 = 0$, and thus the inner problem reads

$$\min_{\|u\|_2 \leq 1} \{ \text{TV}_H(u) + \lambda^k \left( S_2(u) - \langle u, s_1(f^k) \rangle \right) \}. \tag{4}$$

For simplicity we restrict ourselves to submodular balancing functions, in which case $S$ is convex and thus $S_2 = 0$. For the general case, see [16]. Note that the balancing functions of ratio/normalized cut and Cheeger cut are submodular. It turns out that the inner problem is very similar to the semi-supervised learning formulation (3). The efficient solution of both problems is discussed next.

## 5 Algorithms for the Total Variation on Hypergraphs

The problem (3) we want to solve for semi-supervised learning and the inner problem (4) of RatioDCA have a common structure. They are the sum of convex functionals where one of them is the novel regularizer $\Omega_{H,p}$. We propose to solve these problems using a primal-dual algorithm, denoted PDHG in this paper, which was proposed in [23, 24]. Its main idea is to iteratively solve for each convex term in the objective function a so-called proximal problem. Solving the proximal problem w.r.t. a mapping $g : \mathbb{R}^n \to \mathbb{R}$ and a vector $\tilde{x} \in \mathbb{R}^n$ means to compute the proximal map $\text{prox}_g$ defined by

$$\text{prox}_g(\tilde{x}) = \underset{x \in \mathbb{R}^n}{\arg\min} \{ \frac{1}{2} \|x - \tilde{x}\|_2^2 + g(x) \}.$$

The main idea here is that often these proximal problems can be solved efficiently leading to a fast convergence of the overall algorithm. In order to point out the common structure of PDHG for both (3) and the inner problems of Algorithm 1, we first consider a general optimization problem of the form

$$\min_{f \in \mathbb{R}^n} \{ G(f) + F(Kf) \}, \tag{5}$$

where $K \in \mathbb{R}^{m,n}$ and $G : \mathbb{R}^n \to \mathbb{R}$, $F : \mathbb{R}^m \to \mathbb{R}$ are lower-semicontinuous convex functions. Recall that the conjugate function of $G^*$ of $G$ is defined as

$$G^*(x) = \sup_{f \in \mathbb{R}^n} \{ \langle x, f \rangle - G(f) \}$$

and similarly for $F^*$. In terms of these conjugate functions, we can write the dual problem of (5) as

$$-\min_{\alpha \in \mathbb{R}^m} \{ G^*(-K^{\text{T}}\alpha) + F^*(\alpha) \}. \tag{6}$$

The PDHG algorithm for (5) has the following general form. For convergence proofs we refer to [23, 24].

We will now apply this general setting to the convex optimization problems arising in this paper. First, the following Table 1 shows how one can choose $G$ in (5) in order to solve (3) and (4), provides the solutions of the corresponding proximal problems, and gives the conjugate functions. However, note that smooth convex terms can also be directly exploited [25]. Note that we write the constraint in the inner problem of RatioDCA via the indicator function $\iota_{\|\cdot\|_2 \leq 1}$ defined by $\iota_{\|\cdot\|_2 \leq 1}(x) = 0$, if $\|x\|_2 \leq 1$ and $+\infty$ otherwise. Clearly, both proximal problems have an explicit solution.

Second, we discuss the choice of $F$ and $K$ to incorporate $\Omega_{H,p}$.

**Algorithm 2 PDHG**

---

1: **Initialization:** $f^{(0)} = \bar{f}^{(0)} = 0$, $\theta \in [0,1]$, $\sigma, \tau > 0$ with $\sigma\tau < 1/\|K\|_2^2$
2: **repeat**
3:     $\alpha^{(k)} = \text{prox}_{\sigma F^*}(\alpha^{(k)} + \sigma K \bar{f}^{(k)})$
4:     $f^{(k+1)} = \text{prox}_{\tau G(f)}(f^{(k)} - \tau K^{\mathrm{T}}(\alpha^{(k)}))$
5:     $\bar{f}^{(k+1)} = f^{(k+1)} + \theta(f^{(k+1)} - f^{(k)})$
6: **until** relative duality gap $< \epsilon$
7: **Output:** $f^{(k+1)}$.

---

| $G(f) = \frac{1}{2}\|f - Y\|_2^2$ | $G(f) = -\langle s_1(f^k), f \rangle + \iota_{\|\cdot\|_2 \leq 1}(f)$ |
|---|---|
| $\text{prox}_{\tau G(f)}(\tilde{x}) = \frac{1}{1+\tau}(\tilde{x} + \tau Y)$ | $\text{prox}_{\tau G(f)}(\tilde{x}) = \frac{\tilde{x} + \tau s_1(f^k)}{\max\{1, \|\tilde{x} + \tau s_1(f^k)\|_2\}}$ |
| $G^*(x) = \frac{1}{2}\|x + Y\|_2^2 - \frac{1}{2}\|Y\|_2^2$ | $G^*(x) = \|x + s_1(f^k)\|_2$ |

Table 1: Data terms of the SSL functional (3) (left) and the inner problem of RatioDCA (4) (right) with respective proximal map and conjugate.

**PDHG algorithm for $\Omega_{H,1}$.**  Let $m_e$ denote the number of vertices in hyperedge $e \in E$. The main idea is to write

$$\lambda \Omega_{H,1}(f) = F(Kf) := \sum_{e \in E}(F_{(e,1)}(K_e f) + F_{(e,2)}(K_e f)), \tag{7}$$

where the rows of the matrices $K_e \in \mathbb{R}^{m_e, n}$ are the $i$-th standard unit vectors for $i \in e$ and the functionals $F_{(e,j)} : \mathbb{R}^{m_e} \to \mathbb{R}$ are defined as

$$F_{(e,1)}(\alpha^{(e,1)}) = \lambda w_e \max(\alpha^{(e,1)}), \quad F_{(e,2)}(\alpha^{(e,2)}) = -\lambda w_e \min(\alpha^{(e,2)}).$$

The primal problem has thus the form

$$\min_{f \in \mathbb{R}^n}\{G(f) + \sum_{e \in E}(F_{(e,1)}(K_e f) + F_{(e,2)}(K_e f))\}.$$

In contrast to the function $G$, we need in the PDHG algorithm the proximal maps for the conjugate functions of $F_{(e,j)}$. They are given by

$$F^*_{(e,1)} = \iota_{S_{\lambda w_e}}, \quad F^*_{(e,2)} = \iota_{-S_{\lambda w_e}},$$

where $S_{\lambda w_e} = \{x \in \mathbb{R}^{m_e} : \sum_{i=1}^{m_e} x_i = \lambda w_e, x_i \geq 0\}$ is the scaled simplex in $\mathbb{R}^{m_e}$. By (6) the dual problem has the form

$$-\min_{\alpha^{(e,1)}, \alpha^{(e,2)}}\{G^*(-\sum_{e \in E} K_e^{\mathrm{T}}(\alpha^{(e,1)} + \alpha^{(e,2)})) + \sum_{e \in E}(\iota_{S_{\lambda w_e}^e}(\alpha^{(e,1)}) + \iota_{-S_{\lambda w_e}^e}(\alpha^{(e,2)}))\},$$

where $G^*$ is given as in Table 1. The solutions of the proximal problems for $F^*_{(e,1)}$ and $F^*_{(e,1)}$ are the orthogonal projections onto these simplexes written here as $P_{S_{\lambda w_e}^e}$ and $P_{-S_{\lambda w_e}^e}$, respectively. These projections can be performed in linear time, cf., [26].

Using the proximal mappings we have presented so far, we obtain Algorithm 3. In line 1, $c_i = \sum_{e \in E} H_{i,e}$ is the number of hyperedges the vertex $i$ lies in. The bound on the product of the step sizes can be derived as follows

$$\|K\|_2^2 = \|K^{\mathrm{T}}K\|_2 = 2\|\sum_{e \in E} K_e^{\mathrm{T}} K_e\|_2 = 2\max_{i=1,\ldots,n}\{c_i\}.$$

It is important to point out here that the algorithm decouples the problem in the sense that in every iteration we solve subproblems which treat the functionals $G, F_{(e,1)}, F_{(e,2)}$ separately and thus can be solved in an efficient way.

**Algorithm 3 PDHG for $\Omega_{H,1}$**

1: **Initialization:** $f^{(0)} = \bar{f}^{(0)} = 0$, $\theta \in [0,1]$, $\sigma, \tau > 0$ with $\sigma\tau < 1/(2\max_{i=1,\dots,n}\{c_i\})$
2: **repeat**
3: $\quad \alpha^{(e,1)^{(k+1)}} = P_{S_{\lambda w_e}^e}(\alpha^{(e,1)^{(k)}} + \sigma K_e \bar{f}^{(k)}), \quad e \in E$
4: $\quad \alpha^{(e,2)^{(k+1)}} = P_{-S_{\lambda w_e}^e}(\alpha^{(e,2)^{(k)}} + \sigma K_e \bar{f}^{(k)}), \quad e \in E$
5: $\quad f^{(k+1)} = \text{prox}_{\tau G}(f^{(k)} - \tau \sum_{e \in E} K_e^{\mathrm{T}}(\alpha^{(e,1)^{(k+1)}} + \alpha^{(e,2)^{(k+1)}}))$
6: $\quad \bar{f}^{(k+1)} = f^{(k+1)} + \theta(f^{(k+1)} - f^{(k)})$
7: **until** relative duality gap $< \epsilon$
8: **Output:** $f^{(k+1)}$.

**PDHG algorithm for $\Omega_{H,2}$.** We define $G$ and $K_e$ as above. Moreover, we set

$$F_e(\alpha^e) = \lambda w_e \underbrace{(\max(\alpha^e) - \min(\alpha^e))^2}_{=:h_e(\alpha^e)}. \tag{8}$$

Hence, the primal problem can be written as

$$\min_{f \in \mathbb{R}^n}\{G(f) + \sum_{e \in E} F_e(K_e f)\}.$$

In order to formulate the dual problem, we need the conjugate of $F_e$. To this end, we first derive the conjugate function of $h_e$ defined in (8), i.e.,

$$h_e^*(\alpha^e) = \sup_{\phi \in \mathbb{R}^{m_e}} \{\langle \alpha^e, \phi \rangle - (\max(\phi) - \min(\phi))^2\}.$$

**Lemma 5.1.** *Let $\alpha^e \in \mathbb{R}^{m_e}$ and $t_+ = \sum_{i:\alpha_i^e > 0} \alpha_i^e$ and $t_- = \sum_{i:\alpha_i^e < 0} \alpha_i^e$. It holds that*

$$h_e^*(\alpha^e) = \begin{cases} \frac{1}{4}t_+^2 & \text{if } \langle \alpha^e, \mathbf{1} \rangle = 0, \\ +\infty & \text{otherwise.} \end{cases}$$

*Proof.* Using the decomposition, $\phi = \psi + \gamma\mathbf{1}$, where $\langle \psi, \mathbf{1} \rangle = 0$ and $\gamma \in \mathbb{R}$, we can write

$$\langle \alpha^e, \phi \rangle - (\max(\phi) - \min(\phi))^2 = \gamma \langle \alpha^e, \mathbf{1} \rangle + \langle \alpha^e, \psi \rangle - (\max(\psi) - \min(\psi))^2.$$

Thus for $\langle \alpha^e, \mathbf{1} \rangle \neq 0$, we have $h_e^*(\alpha^e) = \infty$. Now we consider the case where $\langle \alpha^e, \mathbf{1} \rangle = 0$. We write $I_- = \{i : \alpha_i^e < 0\}$ and $I_+ = \{i : \alpha_i^e > 0\}$ and define $t_+ = \sum_{i \in I_+} \alpha_i^e$ and $t_- = \sum_{i \in I_-} \alpha_i^e$. Note that $\langle \alpha^e, \mathbf{1} \rangle = 0$ implies $t_+ = -t_-$. Let us assume $a = \max(\phi)$ and $b = \min(\phi)$ are fixed. To maximize $\langle \alpha^e, \phi \rangle - (\max(\phi) - \min(\phi))^2$ it is clearly best to choose $\phi_i = a$ for $i \in I_-$ and $\phi_i = b$ for $i \in I_+$. Consequently,

$$\langle \alpha^e, \phi \rangle - (\max(\phi) - \min(\phi))^2 = t_+(b - a) - (b - a)^2. \tag{9}$$

We maximize the gap $\Delta = b - a$ for the objective $m(\Delta) = t_+\Delta - \Delta^2$ and obtain the maximizer as $\Delta = \frac{t_+}{2}$. Thus we have $h_e^*(\alpha^e) = \frac{t_+^2}{4}$ if $\langle \alpha^e, \mathbf{1} \rangle \neq 0$. $\qquad\square$

With $t_+ = \sum_{i:\alpha_i^e > 0} \alpha_i^e$ and $t_- = \sum_{i:\alpha_i^e < 0} \alpha_i^e$ we thus get

$$F_e^*(\alpha^e) = \lambda w_e h^*\left(\frac{\alpha^e}{\lambda w_e}\right) = \begin{cases} \frac{1}{4\lambda w_e}t_+^2 & \text{if } t_+ = -t_-, \\ +\infty & \text{otherwise.} \end{cases} \tag{10}$$

So, we obtain the dual problem

$$-\min_{\alpha^e}\{G^*(-\sum_{e \in E} K_e^{\mathrm{T}}\alpha^e) + \sum_{e \in E} \frac{1}{4\lambda w_e}(t_+^e)^2 + \sum_{e \in E} \iota_{\{0\}}(t_+^e + t_-^e)\},$$

where $t_+^e = \sum_{i:\alpha_i^e > 0} \alpha_i^e$ and $t_-^e = \sum_{i:\alpha_i^e < 0} \alpha_i^e$.

As we have seen in (10), the conjugate functions $F_e^*$ are not indicator functions and we thus solve the corresponding proximal problems via proximal problems for $F_e$. More specifically, we exploit the fact that

$$\text{prox}_{\sigma F_e^*}(\tilde{\alpha}^e) = \tilde{\alpha}^e - \text{prox}_{\frac{1}{\sigma}F_e}(\tilde{\alpha}^e), \tag{11}$$

see [27, Lemma 2.10], and use the following novel result concerning the proximal problem on the right-hand side of (11).

**Proposition 5.1.** *For any $\sigma > 0$ and any $\tilde{\alpha}^e \in \mathbb{R}^{m_e}$ the proximal map*

$$\text{prox}_{\frac{1}{\sigma}F_e}(\tilde{\alpha}^e) = \underset{\alpha^e \in \mathbb{R}^{m_e}}{\arg\min}\{\frac{1}{2}\|\alpha^e - \tilde{\alpha}^e\|_2^2 + \frac{1}{\sigma}\lambda w_e(\max(\alpha^e) - \min(\alpha^e))^2\}$$

*can be computed with $\mathcal{O}(m_e \log m_e)$ arithmetic operations.*

We will now derive such an algorithm. To simplify the notation, we consider instead of $\frac{1}{\sigma}F_e$ the function $h : \mathbb{R}^m \to \mathbb{R}$ defined by

$$h(\alpha) = (\max(\alpha) - \min(\alpha))^2$$

and show that $\text{prox}_{\mu h}(\alpha)$, $\mu > 0$, can be computed with $\mathcal{O}(m \log m)$ arithmetic operations.

Let us fix $\alpha \in \mathbb{R}^m$. For every pair $r, s \in [\min(\alpha), \max(\alpha)]$ with $r \geq s$, we define $\alpha^{(r,s)}$ by

$$\alpha_i^{(r,s)} = \begin{cases} r & \text{if } \alpha_i \geq r \\ \alpha_i & \text{if } \alpha_i \in (r, s) \\ s & \text{if } \alpha_i \leq s \end{cases} \tag{12}$$

Clearly, if $r = \max(\text{prox}_{\mu h}(\alpha))$ and $s = \min(\text{prox}_{\mu h}(\alpha))$ then $\alpha^{(r,s)} = \text{prox}_{\mu h}(\alpha)$. Hence, the above definition allows us to write the proximal problem in terms of the variables $r, s$ since for

$$(r, s) = \underset{\tilde{r},\tilde{s}}{\arg\min}\{\underbrace{\frac{1}{2}\|\alpha^{(\tilde{r},\tilde{s})} - \alpha\|_2^2}_{=:E_1(\tilde{r},\tilde{s})} + \underbrace{\mu(\tilde{r} - \tilde{s})^2}_{=:E_2(\tilde{r},\tilde{s})}\} \tag{13}$$

we have

$$\text{prox}_{\mu h}(\alpha) = \alpha^{(r,s)}.$$

Our goal is now to find a minimizer of (13). To this end, we first order $\alpha$ in an increasing order which can be done in $\mathcal{O}(m \log m)$ arithmetic operations. W.l.o.g. we assume here that the components of $\alpha$ are pairwise different. Moreover, we introduce the following notation. For $r, s \in [\alpha_1, \alpha_m]$ there exist unique $p, q \in \{1, \ldots, m\}$ characterized by $\alpha_{m-p+1} = \min\{\alpha_i | \alpha_i \geq r\}$ and $\alpha_q = \max\{\alpha_i | \alpha_i \leq s\}$. Thus, the directional partial derivatives w.r.t. $r$ and $s$ are given by

$$\frac{\partial E_1}{\partial r^-}(r, s) = \sum_{i=m-p+1}^{m}(\alpha_i - r), \qquad \frac{\partial E_1}{\partial s^+}(r, s) = \sum_{i=1}^{q}(s - \alpha_i). \tag{14}$$

They tell us how much we increase $E_1$ by decreasing $r$ and increasing $s$, respectively. On the other hand both of these changes lead to a decrease in the energy $E_2$. More precisely, it holds that

$$\frac{\partial E_2}{\partial r^-}(r, s) = \frac{\partial E_2}{\partial s^+}(r, s) = 2\mu(s - r). \tag{15}$$

Thus, the main ideas behind our algorithm are as follows. Starting with $r = \max(\alpha)$ and $s = \min(\alpha)$, we decrease $r$ and increase $s$ keeping the two partial derivatives of (14) equal. We stop when the sum of the partial derivatives vanishes. So, the optimal $r, s$ are characterized by the system

$$\sum_{i=m-p+1}^{m}(\alpha_i - r) = \sum_{i=1}^{q}(s - \alpha_i), \tag{16}$$

$$\sum_{i=m-p+1}^{m}(\alpha_i - r) + 2\mu(s - r) = 0. \tag{17}$$

We will now generate a sequence of pairs $r^{(k)}, s^{(k)}$ satisfying $r^{(k)} \geq s^{(k)}$ and (16) for each $k$. The corresponding indices needed to calculate the partial derivatives will be denoted by $p^{(k)}, q^{(k)}$. The main procedure is described in the next lemma.

**Lemma 5.2.** *Assume* $r^{(k)} \in (\alpha_{m-p^{(k)}}, \alpha_{m-p^{(k)}+1}]$ *and* $s^{(k)} \in [\alpha_{q^{(k)}}, \alpha_{q^{(k)}+1})$ *and property* (16) *holds for* $(r^{(k)}, s^{(k)})$. *Then, we can either choose*

$$r^{(k+1)} = r^{(k)} - \frac{q^{(k)}}{p^{(k)}}(s^{(k+1)} - s^{(k)}) \ \ and \ \ s^{(k+1)} = \alpha_{q^{(k)}+1} \tag{18}$$

*or*

$$r^{(k+1)} = \alpha_{m-p^{(k)}} \ \ and \ \ s^{(k+1)} = s^{(k)} + \frac{p^{(k)}}{q^{(k)}}(r^{(k)} - r^{(k+1)}) \tag{19}$$

*such that* $r^{(k+1)} \in [\alpha_{m-p^{(k)}}, \alpha_{m-p^{(k)}+1})$, $s^{(k+1)} \in (\alpha_{q^{(k)}}, \alpha_{q^{(k)}+1}]$ *and* (16) *holds true for* $(r^{(k+1)}, s^{(k+1)})$.

*Proof.* Property (16) for $(r^{(k+1)}, s^{(k+1)})$ means that

$$\sum_{i=m-p^{(k)}+1}^{m} (\alpha_i - r^{(k+1)}) = \sum_{i=1}^{q^{(k)}} (s^{(k+1)} - \alpha_i). \tag{20}$$

Since by assumption (16) holds for $(r^{(k)}, s^{(k)})$, equation (20) is equivalent to

$$p^{(k)}(r^{(k+1)} - r^{(k)}) = q^{(k)}(s^{(k)} - s^{(k+1)}).$$

If we set $(r^{(k+1)}, s^{(k+1)})$ according to (18) but $r^{(k+1)} < \alpha_{m-p^{(k)}}$. Then we get

$$r^{(k)} - \frac{q^{(k)}}{p^{(k)}}(\alpha_{q^{(k)}+1} - s^{(k)}) < \alpha_{m-p^{(k)}}$$

$$\Rightarrow s^{(k)} + \frac{p^{(k)}}{q^{(k)}}(r^{(k)} - \alpha_{m-p^{(k)}}) < \alpha_{q^{(k)}+1},$$

i.e., we can choose $r^{(k+1)}, s^{(k+1)}$ according to (19) and vice versa. $\square$

After each computation of a new pair $(r^{(k+1)}, s^{(k+1)})$ we check if the left-hand side of (17) is smaller than zero (note that initially the left-hand side of (17) is negative and it is increasing for every iteration). If this is not the case, we found the intervals where the optimal values $r$ and $s$ lie in. Restricted to this domain the functional $E_1 + E_2$ is a differentiable. Hence, we can compute $r, s$ as follows.

**Lemma 5.3.** *Assume that the optimal* $r, s$ *of* (13) *fulfill* $r \in [\alpha_{m-p}, \alpha_{m-p+1}]$ *and* $s \in [\alpha_q, \alpha_{q+1}]$. *Then, it holds that*

$$s = \left(q + 2\mu - \frac{(2\mu)^2}{\sum_{i=m-p+1}^{m}\alpha_i + 2\mu}\right)^{-1}\left(\frac{2\mu}{p+2\mu}\sum_{i=m-p+1}^{m}\alpha_i + \sum_{i=1}^{q}\alpha_i\right)$$

$$r = \frac{1}{2\mu}\left((q+2\mu)s - \sum_{i=1}^{q}\alpha_i\right).$$

*Proof.* When restricted to $[\alpha_i, \alpha_{i+1}] \times [\alpha_j, \alpha_{j+1}]$, the function $(r, s) \mapsto E_1(r, s) + E_2(r, s)$ is a quadratic function in $(r, s)$. We can thus simply set the gradient to zero and solve the corresponding system of linear equations which yields the above result. $\square$

In conclusion, we obtain the following algorithm. Note that after the sorting, the algorithm takes in the order of $m$ steps to compute the proximal map which proves Proposition 5.1.

Hence, the corresponding PDHG algorithm can be formulated as follows.

We solve the subproblems in line 3 via Algorithm 4. Note that the bound on the step sizes is now doubled, i.e., less restrictive since we have defined for each hyperedge one functional $F_e$ and not two as for $p = 1$, i.e.,

$$\|K\|_2^2 = \|K^{\mathsf{T}}K\|_2 = \|\sum_{e \in E} K_e^{\mathsf{T}}K_e\|_2 = \max_{i=1,\dots,n}\{c_i\}.$$

---

**Algorithm 4** – Solution of the proximal problem $\mathrm{prox}_{\mu h}(\alpha)$

---

1: Sort $\alpha \in \mathbb{R}^m$ in increasing order.
2: **Initialization:** $r^{(0)} = \max(\alpha), s^{(0)} = \min(\alpha)$
3: **while** $\frac{\partial E_1}{\partial r^-}(r^{(k)}, s^{(k)}) < 2\mu(r^{(k)} - s^{(k)})$ and $q^{(k)} + 1 \leq m - p^{(k)}$ **do**
4:     Find $(r^{(k+1)}, s^{(k+1)})$ according to Lemma 5.2.
5: **end while**
6: Compute $r, s$ as described in Lemma 5.3.
7: **Output:** After restoring the original order, set

$$(\mathrm{prox}_{\mu h}(\alpha))_i = \left\{ \begin{array}{ll} r & \text{if } \alpha_i \geq r, \\ \alpha_i & \text{if } \alpha_i \in (r, s), \\ s & \text{if } \alpha_i \leq s, \end{array} \right. \text{ for } i = 1, \ldots, m.$$

---

---

**Algorithm 5 PDHG for $\Omega_{H,2}$**

---

1: **Initialization:** $f^{(0)} = \bar{f}^{(0)} = 0$, $\theta \in [0,1]$, $\sigma, \tau > 0$ with $\sigma\tau < 1/\max_{i=1,\ldots,n}\{c_i\}$
2: **repeat**
3:     $\alpha^{e(k+1)} = \alpha^{e(k)} + \sigma K_e \bar{f}^{(k)} - \mathrm{prox}_{\frac{1}{\sigma}F_e}(\alpha^{e(k)} + \sigma K_e \bar{f}^{(k)}), \quad e \in E$
4:     $f^{(k+1)} = \mathrm{prox}_{\tau G}(f^{(k)} - \tau \sum_{e \in E} K_e^{\mathrm{T}}(\alpha^{e(k+1)}))$
5:     $\bar{f}^{(k+1)} = f^{(k+1)} + \theta(f^{(k+1)} - f^{(k)})$
6: **until** relative duality gap $< \epsilon$
7: **Output:** $f^{(k+1)}$.

---

## 6 Experiments

The method of Zhou et al [11] seems to be the standard algorithm for clustering and SSL on hypergraphs. We compare to them on a selection of UCI datasets summarized in Table 2. Zoo, Mushrooms and 20Newsgroups[2] have been used also in [11] and contain only categorical features. As in [11], a hyperedge of weight one is created by all data points which have the same value of a categorical feature. For covertype we quantize the numerical features into 10 bins of equal size. Two datasets are created each with two classes (4,5 and 6,7) of the original dataset.

**Semi-supervised Learning (SSL).** In [11], they suggest using a regularizer induced by the normalized Laplacian $L_{CE}$ arising from the clique expansion

$$L_{CE} = \mathbb{I} - D_{CE}^{-\frac{1}{2}} H W' H^T D_{CE}^{-\frac{1}{2}},$$

where $D_{CE}$ is a diagonal matrix with entries $d_{EC}(i) = \sum_{e \in E} H_{i,e} \frac{w_e}{|e|}$ and $W' \in \mathbb{R}^{|E| \times |E|}$ is a diagonal matrix with entries $w'(e) = w_e/|e|$. The SSL problem can then be formulated as

$$\lambda > 0, \qquad \arg\min_{f \in \mathbb{R}^{|V|}} \{\|f - Y\|_2^2 + \lambda \langle f, L_{CE} f \rangle\}.$$

| Prop. \ Dataset | Zoo | Mushrooms | Covertype (4,5) | Covertype (6,7) | 20Newsgroups |
|---|---|---|---|---|---|
| Number of classes | 7 | 2 | 2 | 2 | 4 |
| $|V|$ | 101 | 8124 | 12240 | 37877 | 16242 |
| $|E|$ | 42 | 112 | 104 | 123 | 100 |
| $\sum_{e \in E} |e|$ | 1717 | 170604 | 146880 | 454522 | 65451 |
| $|E|$ of Clique Exp. | 10201 | 65999376 | 143008092 | 1348219153 | 53284642 |

Table 2: Datasets used for SSL and clustering. Note that the clique expansion leads for all datasets to a graph which is close to being fully connected as all datasets contain large hyperedges. For covertype (6,7) the weight matrix needs over 10GB of memory, the original hypergraph only 4MB.

The advantage of this formulation is that the solution can be found via a linear system. However, as Table 2 indicates the obvious downside is that $L_{CE}$ is a potentially very dense matrix and thus one needs in the worst case $|V|^2$ memory and $O(|V|^3)$ computations. This is in contrast to our method which needs $2\sum_{e\in E}|e|+|V|$ memory. For the largest example (covertype 6,7), where the clique expansion fails due to memory problems, our method takes 30-100s (depending on $\lambda$). We stop our method for all experiments when we achieve a relative duality gap of $10^{-6}$. In the experiments we do 10 trials for different numbers of labeled points. The reg. parameter $\lambda$ is chosen for both methods from the set $10^{-k}$, where $k=\{0,1,2,3,4,5,6\}$ via 5-fold cross validation. The resulting errors and standard deviations can be found in the following table(first row lists the no. of labeled points).

Our SSL methods based on $\Omega_{H,p}$, $p=1,2$ outperform consistently the clique expansion technique of Zhou et al [11] on all datasets except 20newsgroups[3]. However, 20newsgroups is a very difficult dataset as only 10,267 out of the 16,242 data points are different which leads to a minimum possible error of $9.6\%$. A method based on pairwise interaction such as the clique expansion can better deal with such label noise as the large hyperedges for this dataset accumulate the label noise. On all other datasets we observe that incorporating hypergraph structure leads to much better results. As expected our squared TV functional ($p=2$) outperforms slightly the total variation ($p=1$) even though the difference is small. Thus, as $\Omega_{H,2}$ reduces to the standard regularization based on the graph Laplacian, which is known to work well, we recommend $\Omega_{H,2}$ for SSL on hypergraphs.

| Zoo | 20 | 25 | 30 | 35 | 40 | 45 | 50 |
|---|---|---|---|---|---|---|---|
| Zhou et al. | $35.1\pm17.2$ | $30.3\pm7.9$ | $40.7\pm14.2$ | $29.7\pm8.8$ | $32.9\pm16.8$ | $27.6\pm10.8$ | $25.3\pm14.4$ |
| $\Omega_{H,1}$ | $2.9\pm3.0$ | $\mathbf{1.4\pm2.2}$ | $\mathbf{2.2\pm2.1}$ | $\mathbf{0.7\pm1.0}$ | $\mathbf{0.7\pm1.5}$ | $\mathbf{0.9\pm1.4}$ | $1.9\pm3.0$ |
| $\Omega_{H,2}$ | $\mathbf{2.3\pm1.9}$ | $1.5\pm2.4$ | $2.9\pm2.3$ | $0.9\pm1.4$ | $0.8\pm1.7$ | $1.2\pm1.8$ | $\mathbf{1.6\pm2.9}$ |

| Mushr. | 20 | 40 | 60 | 80 | 100 | 120 | 160 | 200 |
|---|---|---|---|---|---|---|---|---|
| Zhou et al. | $\mathbf{15.5\pm12.8}$ | $10.9\pm4.4$ | $9.5\pm2.7$ | $10.3\pm2.0$ | $9.0\pm4.5$ | $8.8\pm1.4$ | $8.8\pm2.3$ | $9.3\pm1.0$ |
| $\Omega_{H,1}$ | $19.5\pm10.5$ | $10.8\pm3.7$ | $\mathbf{7.4\pm3.8}$ | $\mathbf{5.6\pm1.9}$ | $\mathbf{5.7\pm2.2}$ | $5.4\pm2.4$ | $4.9\pm3.8$ | $5.6\pm3.8$ |
| $\Omega_{H,2}$ | $18.4\pm7.4$ | $\mathbf{9.8\pm4.5}$ | $9.9\pm5.5$ | $6.4\pm2.7$ | $6.3\pm2.5$ | $\mathbf{4.5\pm1.8}$ | $\mathbf{4.4\pm2.1}$ | $\mathbf{3.0\pm0.6}$ |

| covert45 | 20 | 40 | 60 | 80 | 100 | 120 | 160 | 200 |
|---|---|---|---|---|---|---|---|---|
| Zhou et al. | $\mathbf{18.9\pm4.6}$ | $18.3\pm5.2$ | $17.2\pm6.7$ | $16.6\pm6.4$ | $17.6\pm5.2$ | $18.4\pm5.1$ | $19.2\pm4.0$ | $20.4\pm2.9$ |
| $\Omega_{H,1}$ | $21.4\pm0.9$ | $17.6\pm2.6$ | $12.6\pm4.3$ | $7.6\pm3.5$ | $6.2\pm3.8$ | $4.5\pm3.6$ | $2.6\pm1.6$ | $1.5\pm1.3$ |
| $\Omega_{H,2}$ | $20.7\pm2.0$ | $\mathbf{16.1\pm4.1}$ | $\mathbf{10.9\pm4.9}$ | $\mathbf{5.9\pm3.7}$ | $\mathbf{4.6\pm3.4}$ | $\mathbf{3.3\pm3.1}$ | $\mathbf{2.2\pm1.8}$ | $\mathbf{1.0\pm1.1}$ |

| covert67 | 20 | 40 | 60 | 80 | 100 | 120 | 160 | 200 |
|---|---|---|---|---|---|---|---|---|
| $\Omega_{H,1}$ | $40.6\pm8.9$ | $6.4\pm10.4$ | $3.6\pm3.2$ | $3.3\pm2.5$ | $1.8\pm0.8$ | $1.3\pm0.9$ | $0.9\pm0.4$ | $1.2\pm0.9$ |
| $\Omega_{H,2}$ | $\mathbf{25.2\pm18.3}$ | $\mathbf{4.3\pm9.6}$ | $\mathbf{2.1\pm2.0}$ | $\mathbf{2.2\pm1.4}$ | $\mathbf{1.4\pm1.1}$ | $\mathbf{1.0\pm0.8}$ | $\mathbf{0.7\pm0.4}$ | $\mathbf{1.1\pm0.8}$ |

| 20news | 20 | 40 | 60 | 80 | 100 | 120 | 160 | 200 |
|---|---|---|---|---|---|---|---|---|
| Zhou et al. | $\mathbf{45.5\pm7.5}$ | $\mathbf{34.4\pm3.1}$ | $\mathbf{31.5\pm1.4}$ | $\mathbf{29.8\pm4.0}$ | $\mathbf{27.0\pm1.3}$ | $\mathbf{27.3\pm1.5}$ | $\mathbf{25.7\pm1.4}$ | $\mathbf{25.0\pm1.3}$ |
| $\Omega_{H,1}$ | $65.7\pm6.1$ | $61.4\pm6.1$ | $53.2\pm5.7$ | $46.2\pm3.7$ | $42.4\pm3.3$ | $40.9\pm3.2$ | $36.1\pm1.5$ | $34.7\pm3.6$ |
| $\Omega_{H,2}$ | $55.0\pm4.8$ | $48.0\pm6.0$ | $45.0\pm5.9$ | $40.3\pm3.0$ | $38.3\pm2.7$ | $38.1\pm2.6$ | $35.0\pm2.8$ | $34.1\pm2.0$ |

Test error and standard deviation of the SSL methods over 10 runs for varying number of labeled points.

**Clustering.** We use the normalized hypergraph cut as clustering objective. For more than two clusters we recursively partition the hypergraph until the desired number of clusters is reached. For comparison we use the normalized spectral clustering approach based on the Laplacian $L_{CE}$ [11](clique expansion). The first part (first 6 columns) of the following table shows the clustering errors (majority vote on each cluster) of both methods as well as the normalized cuts achieved by these methods on the hypergraph and on the graph resulting from the clique expansion. Moreover, we show results (last 4 columns) which are obtained based on a $k$NN graph (unit weights) which is built based on the Hamming distance (note that we have categorical features) in order to check if the hypergraph modeling of the problem is actually useful compared to a standard similarity based graph construction. The number $k$ is chosen as the smallest number for which the graph becomes connected and we compare results of normalized 1-spectral clustering [14] and the standard spectral clustering [22]. Note that the employed hypergraph construction has no free parameter.

| Dataset | Clustering Error % Ours | [11] | Hypergraph Ncut Ours | [11] | Graph(CE) Ncut Ours | [11] | Clustering Error % [14] | [22] | $k$NN-Graph Ncut [14] | [22] |
|---|---|---|---|---|---|---|---|---|---|---|
| Mushrooms | 10.98 | 32.25 | 0.0011 | 0.0013 | 0.6991 | 0.7053 | 48.2 | 48.2 | 1e-4 | 1e-4 |
| Zoo | 16.83 | 15.84 | 0.6739 | 0.6784 | 5.1315 | 5.1703 | 5.94 | 5.94 | 1.636 | 1.636 |
| 20-newsgroup | 47.77 | 33.20 | 0.0176 | 0.0303 | 2.3846 | 1.8492 | 66.38 | 66.38 | 0.1031 | 0.1034 |
| covertype (4,5) | 22.44 | 22.44 | 0.0018 | 0.0022 | 0.7400 | 0.6691 | 22.44 | 22.44 | 0.0152 | 0.02182 |
| covertype (6,7) | 8.16 | - | 8.18e-4 | - | 0.6882 | - | 45.85 | 45.85 | 0.0041 | 0.0041 |

First, we observe that our approach optimizing the normalized hypergraph cut yields better or similar results in terms of clustering errors compared to the clique expansion (except for 20-newsgroup for the same reason given in the previous paragraph). The improvement is significant in case of Mushrooms while for Zoo our clustering error is slightly higher. However, we always achieve smaller normalized hypergraph cuts. Moreover, our method sometimes has even smaller cuts on the graphs resulting from the clique expansion, although it does not directly optimize this objective in contrast to [11]. Again, we could not run the method of [11] on covertype (6,7) since the weight matrix is very dense. Second, the comparison to a standard graph-based approach where the similarity structure is obtained using the Hamming distance on the categorical features shows that using hypergraph structure is indeed useful. Nevertheless, we think that there is room for improvement regarding the construction of the hypergraph which is a topic for future research.

### Acknowledgments

M.H. would like to acknowledge support by the ERC Starting Grant NOLEPRO and L.J. acknowledges support by the DFG SPP-1324.

## Footnotes

[1]A function $f : \mathbb{R}^d \to \mathbb{R}$ is positively 1-homogeneous if $\forall \alpha > 0$, $f(\alpha x) = \alpha f(x)$.

[2]This is a modified version by Sam Roweis of the original 20 newsgroups dataset available at http://www.cs.nyu.edu/~roweis/data/20news_w100.mat.

[3]Communications with the authors of [11] could not clarify the difference to their results on 20newsgroups