[Reviews · NeurIPS 2013]

Submitted by Assigned_Reviewer_4

This paper proposes an algorithm for normalized cuts of hypergraphs by
formulating the cut as the minimization of a ratio of two convex
functions which can be solved using existing methods (RatioDCA, with
an inner problem solved using a primal-dual method). Semi-supervised
learning on a hypergraph is formulated as a related optimization
problem and solved with a similar primal-dual method. The proposed
approach is shown on several datasets to outperform an alternative
technique based on a transformation of the hypergraph to a regular
graph for a semi-supervised learning, a clustering and a cut
objective.

The paper is clear and well written. It is technically sound and
provides a significant contribution to the problem of hypergraph cut,
and possibly to semi-supervised learning and clustering --- assuming a
hypergraph based approach is relevant to the problem.

Concerning this last point, not much is said about the relevance of
the hypergraph approach. In all examples, the hypergraph is not
provided as a separate structure, but is built from the covariates by
making one hyperedge for all samples which share the same value of one
covariate. if the data points are originally represented as vectors of
features, other semi-supervised and clustering techniques (eg based on
scalar products or on a regular graph built from the features) would
make sense and should be compared against.

Similarly, the relevance of semi-supervised learning to the datasets
is not discussed: what kind of performance would be obtained without
using the unlabeled samples? Admittedly, the point of this paper is
how the new formulation improves semi-supervised, not whether
semi-supervised is relevant.

A few minor points:

- Total variation seems to be accessory in the paper, whose main
achievement is to provide a better algorithm for hypergraph
normalized cuts. The current title is a little misleading from this
point of view.

- The semi-supervised problem (3) is itself a prox of \lambda\Omega at
point Y. There may not be a simpler way to compute the prox than the
proposed algorithm but it could be useful to point it out in order
to avoid confusion between prox (3) and the prox of its first term
and the conjugate of the second term which are used to compute it.
Summary: - Clear presentation, well written paper.
- Significant contribution, technically sound.
- Little discussion of the relevance of using hypergraphs to represent
this data (as opposed to vectors or graphs).

Submitted by Assigned_Reviewer_5

Graph cut methods for semi-supervised classification and clustering are dominant in the last decade. Hypergraphs can incorporate higher order information about data than ordinary graphs and thus should be more preferable. Existing methods all have their own sets of limitation, as discusses in Section 1.

In contrast, this paper studies how to directly deal with the hypergraph cut using the total variation of the Lovasz extension. Two frameworks are proposed in Sections 3 and 4, and an algorithm for solving the involved problems is presented in Section 5. Experimental results in Section 6 are promising.

The idea of this paper sounds quite interesting. However, the derivation is not easy to follow. I have not checked the technical details since I am not familiar with these optimization problems in Sections 4 and 5. Anyway, the extension in Section 3 for semi-supervised learning from graph Laplacian matrices to the proposed functionals is natural, and the theoretical results seem correct.

I have two minor questions. Firstly, in Definition 2.1, it is required that f_1<=...<=f_n. It seems later in the optimizations f as an optimization variable may not always satisfy this constraint. So when is it valid? Secondly, in the experiments, numerical features are converted into categorical by 10 equal size bins. Will this cause certain loss of information?

** Comment after authors' feedback **

The authors have clarified my concern about the experimental setup.
Summary: This paper studies how to directly deal with the hypergraph cut using the total variation of the Lovasz extension. The idea is quite interesting, and the experimental results are promising.

Submitted by Assigned_Reviewer_8

The paper defines a notion of total variation on hypergraphs, proves its convexity and relation to cuts, discusses applications to learning and normalized cuts (for clustering), computes proximity operators as part of optimization algorithms, and presents numerical experiments comparing with a popular method of Zhou from 2006. Overall, the paper tours several sub-disciplines. There are a lot of ideas and a few theorems (though of course they cannot say much about a global solution in the case of the normalized cut problem, but no one can). The paper stays focused despite the amount of material and it is well written, with only occasional grammar issues.

Section 5 falls most into my area of expertise, and I am satisfied with it. I am less familiar with existing work on spectral clustering and existing approaches for hypergraphs. This paper is basically claiming to be the first major advance in 7 years (since Zhou [11]), and I am inclined to believe this, but not completely certain. If it is true, then this paper should definitely be accepted and will likely become well-known. But of course I am a bit skeptical that the authors have not found any more recent work than [11], given the amount of attention to the subject (or is it because everyone is working on tensors and ignoring hypergraphs?)

Other comments:
- Comparisons were always with [11], which uses the same hypergraph framework, and then reduces it with the CE technique to a graph problem. But these problems (SSL, clustering) can be attacked from quite different perspectives. A comparison with an "outside" approach would make this paper stronger, even a simple algorithm (such as k-means for clustering... or explain why it is not applicable). Your clustering approach which recursively splits clusters is clearly not optimal (though I do not doubt that it is a standard technique).

- Line 121: "the optimal cut". Shouldn't this be "an optimal cut", since it is symmetric so and the mirror-image cut would work as well?

- The section around Thm 4.1 seemed vague to me and could have benefited from more explanation. For example, it's not clear to me how to apply the second equation from the theorem.

- I'm not sure what the authors mean by "tight relaxation" (section 4, and mentioned in the intro). Is it the "the tightest"? (in that case, define what is your class of "relaxations", since it's not the class of convex functions in this case). Or do you mean "tighter" than the linear eigenproblem relaxation? (in this case, how do you justify this? For graphs, it is justified by empirical evidence, so for hypergraphs, you should derive the linear relaxation and test it).

- Section 5. The method in [24] has since been extended a lot; see, for example, Condat 2011 (http://www.optimization-online.org/DB_HTML/2011/12/3284.html). In particular (and I think this was already in [24]), it can exploit smooth functions. Usually, it is better (in terms of convergence rates) to take derivatives of smooth functions when possible, as opposed to calculating their proximity operators; so for the first G term in Table 1, why not treat it as the smooth term? For line 276, "the main idea" (also, note that this phrase is repeated three times on this page) is not really described that well, since this is not unique to the method of [24]. Rather [24] allows one to separate many terms.

- I am quite familiar with TV on a grid (for 2D). In this case, people prefer using the isotropic TV, but this does not have a closed-form proximity operator nor an efficient algorithm to compute it. Alternatively, the most common form of anisotropic TV is much simpler to work with. It appears that your TV definition generalizes this anisotropic TV. It's not clear, but it would be very interesting to explore, if it is possible to define an isotropic TV on a hypergraph.

- Prop 5.1. This is O( n log n) due to the sort. Is it not possible to avoid the sort and just find the largest and smallest entries of the input? This would bring it down to O(n). I realize you may have to worry about cases when the answer will have several values at the max/min, but it still may be possible to deal with this.

- Experiments. Overall, well done. For making the hypergraph edges using data points that have the same value of a feature, is it not beneficial to make edges (but with much lower weight) when data points have *similar* values?

- The table on page 8: what are the entries? Some kind of error? This was not explained

- Line 415: "Our method... minimizes the normalized cut on the hypergraph." Unless I have misunderstood, this is not true. You apply a convex method to a nonconvex problem, so in the best case (and this you would need to show) you reach a stationary point or local minima. So at least say that your method "attempts" to minimize...

Grammar:
- Line 276 and 301, "proximum" and "proxima" are not commonly used; just use "proximity operator".

- Line 416-7 is awkward.

- The authors use the construction "allow to" a lot, as well as variants "suggest/recommend to use", "favor to split off", "propose to solve", (e.g., line 17, 69, 107, 192, 201, 385). Since the English is otherwise nearly perfect, it's worth correcting this. In these cases, the infinite should be a gerund, e.g. "suggest to use" should be "suggest using". Rarely, both forms are correct (e.g. "I like to play tennis" and "I like playing tennis" are both fine). In the case of "Hypergraphs allow to encode", it should be "Hypergraphs allow one to encode".

- "However" is not used correctly a few times. E.g. line 199, it should be "Hence" not "However", and line 408 it should be "Thus".
Summary: This seems like a significant paper. It introduces a new topic and explores it quite a bit, with good results. The numerics are done well, and it compares quite favorably over a similar approach from 2006. My only reservation is whether they have missed some recent work from the past 7 years that has further explored hypergraphs and would improve on [11].
Author Feedback

Author rebuttal: We thank all reviewers for their suggestions and comments.
We will incorporate them into the final version.

Please note that we answer the questions and comments in a thematic order.


Reviewer 1 - Relevance of the hypergraph approach

The main goal of the experiments is to compare to the method
of Zhou et al [11]. Thus we basically repeated the experiments
in the same setup (datasets and hypergraph construction). However, the relevance of hypergraphs has been
pointed out, e.g., for motion segmentation [2], subspace clustering [9]
and analysis of cellular networks [3]. We agree with the reviewer
that for the chosen datasets other clustering approaches would be
possible - we are not claiming that this is the best possible way
to solve these clustering problems. If space allows we will try to
incorporate one of the above problems in the final version.


Reviewer 2 - 'in Definition 2.1, it is required that f_1<=...<=f_n.
It seems later in the optimizations f as an optimization variable may not
always satisfy this constraint.'

We are abusing notation a bit in the definition of the Lovasz extension. What we mean is the following: Let pi be a permutation of {1,...,|V|} such that f_{pi_1} <= f_{pi_2} <= ... <= f_{pi_n}, then replace in the definition of S(f)
f_i by f_{pi_i}. In the algorithm, this does not play a role
because we derive an analytical form of the Lovasz extension of the hypergraph
cut in Proposition 2.2. Hence, there is no need for an additional ordering of the input vector.


Reviewer 2 - 'numerical features are converted into categorical by 10 equal size bins.
Will this cause certain loss of information?'

Yes, even though one can argue that for this particular dataset the loss is small.
However, we are following the experiment design of [11] in order to have a fair comparison (please see the above comment to Reviewer 1). We are not arguing that this is necessarily the best way to solve these clustering problems. However,
the hypergraph approach definitely makes sense for categorical features.


Reviewer 3 - 'This paper is basically claiming to be the first major advance in 7 years (since Zhou [11])'

In the meantime there have been advances in the tensor approach to tackle k-uniform hypergraphs
e.g. [9] is from AISTATS 2012. But for general hypergraphs we have found after an extensive
literature review no significant step after [10,11,12] e.g. paper [2] from CVPR 2012 uses [11]
but modifies the weight definition of the cliques.


Reviewer 3 - 'A comparison with an "outside" approach would make this paper stronger, even a simple algorithm
(such as k-means for clustering... or explain why it is not applicable)'

Please see our comments to Reviewer 1 and Reviewer 2 - we will include a comparison to other clustering methods but k-means
cannot be applied in a straightforward way to categorical features


Reviewer 3 - 'Your clustering approach which recursively splits clusters is clearly not optimal '

It is not optimal but it is the standard way to get a multi-cut.


Reviewer 3 - 'The section around Thm. 4.1 seemed vague to me and could have benefited from more explanation.
For example, it's not clear to me how to apply the second equation from the theorem.'
'I'm not sure what the authors mean by "tight relaxation"'

Thm 4.1. states an exact relaxation of the balanced hypergraph cut problem, that is the
discrete combinatorial problem is equivalent to the continuous optimization problem stated
in Thm 4.1. (in this sense tight).
The inequality in Thm 4.1. is very useful as finally for the clustering we need a way
to go from the continuous problem back to the combinatorial one. This is done by
thresholding: Thm 4.1. says that thresholding of the continuous solution can only
improve the balanced graph cut. Note that for f=1_C the continuous objective equals
cut(C,\overline{C})/\hat{S}(C).


Reviewer 3 - Better optimization techniques beyond [23,24]

We are grateful to the reviewer for making us aware of the recent improvements
in primal-dual methods. We will explore this for further speed-up.


Reviewer 3 - 'is it possible to define an isotropic TV on a hypergraph'

As isotropy is directly related to invariances with respect to the Euclidean
group, we doubt that this can be generalized to hypergraphs.


Reviewer 3 - 'Prop 5.1. This is O( n log n) due to the sort. Is it not possible to avoid the sort and just find the largest and smallest entries of the input?'

We are currently exploring techniques to improve it to O(n).


Reviewer 3 - 'is it not beneficial to make edges (but with much lower weight) when data points have *similar* values'

In general yes, but note that for most datasets all features are categorical - then one needs a notion of similarity
on these features to do this.


Reviewer 3 - 'The table on page 8: what are the entries? Some kind of error? This was not explained'

It shows the error+standard deviation (10 runs) for different number of labeled points. We will make this clearer in the final version.


Reviewer 3 - 'Our method... minimizes the normalized cut on the hypergraph." Unless I have misunderstood, this is not true. You apply a convex method to a nonconvex problem, so in the best case (and this you would need to show) you reach a stationary point or local minima. So at least say that your method "attempts" to minimize...'

We will clarify this. We wanted to emphasize again the difference between minimizing normalized hypergraph cut
and normalized graph cut for the clique expansion but we are clearly not claiming to have guarantees to find
the global optimum.


Reviewer 3 - Grammar mistakes

As non-native speakers we are very grateful for such hints - Thanks a lot !